# Distinct dendritic Ca²⁺ spike forms produce opposing input-output transformations in rat CA3 pyramidal cells

**Ádám Magó[1†], Noémi Kis[1,2†], Balázs Lükő[1], Judit K Makara[1]\***

[1]Laboratory of Neuronal Signaling, Institute of Experimental Medicine, Budapest, Hungary; [2]János Szentágothai School of Neurosciences, Semmelweis University, Budapest, Hungary

**\*For correspondence:**
makara.judit@koki.hu

[†]These authors contributed equally to this work

**Competing interest:** The authors declare that no competing interests exist.

**Abstract** Proper integration of different inputs targeting the dendritic tree of CA3 pyramidal cells (CA3PCs) is critical for associative learning and recall. Dendritic Ca²⁺ spikes have been proposed to perform associative computations in other PC types by detecting conjunctive activation of different afferent input pathways, initiating afterdepolarization (ADP), and triggering burst firing. Implementation of such operations fundamentally depends on the actual biophysical properties of dendritic Ca²⁺ spikes; yet little is known about these properties in dendrites of CA3PCs. Using dendritic patch-clamp recordings and two-photon Ca²⁺ imaging in acute slices from male rats, we report that, unlike CA1PCs, distal apical trunk dendrites of CA3PCs exhibit distinct forms of dendritic Ca²⁺ spikes. Besides ADP-type global Ca²⁺ spikes, a majority of dendrites expresses a novel, fast Ca²⁺ spike type that is initiated locally without bAPs, can recruit additional Na⁺ currents, and is compartmentalized to the activated dendritic subtree. Occurrence of the different Ca²⁺ spike types correlates with dendritic structure, indicating morpho-functional heterogeneity among CA3PCs. Importantly, ADPs and dendritically initiated spikes produce opposing somatic output: bursts versus strictly single-action potentials, respectively. The uncovered variability of dendritic Ca²⁺ spikes may underlie heterogeneous input-output transformation and bursting properties of CA3PCs, and might specifically contribute to key associative and non-associative computations performed by the CA3 network.

## Editor's evaluation

In this technically challenging study of CA3 pyramidal neurons in adult rats, involving dendritic patch-clamp electrophysiology from thin distal dendrites, the authors systematically and rigorously characterize dendritic calcium spikes and their heterogeneities in this neuronal subtype. The authors used dual somatic-dendritic recording, two-photon calcium imaging, and targeted glutamate uncaging to explore the active properties of dendrites of CA3 pyramidal neurons from rat hippocampus. They discovered two fundamentally different types of regenerative calcium events or spikes in the dendrites of these cells. One remained fairly local while the other propagated widely throughout the dendrites. The authors conclude that these different calcium spikes contribute to various single neuron computations. The authors rightly emphasize cell-type dependence of dendritic physiology across different neuronal subtypes and underscore the need to account for these differences in understanding single-neuron and circuit physiology. Overall, this interesting, important, and well-done study delineates the different dendritic calcium events in CA3 pyramidal neurons employing various kinds of stimuli.

## Introduction

Dendrites play a critical role in the integration and plasticity of synaptic inputs. Voltage-dependent ion channels and passive electrical properties of dendrites enable neurons to perform various forms of linear and nonlinear input-output transformation. In particular, cortical pyramidal cell (PC) dendrites are thought to support multiple types of regenerative dendritic spikes (d-spikes), including Na$^+$ spikes (mediated by voltage-gated Na$^+$ channels [VGNCs]), NMDA spikes (mediated by NMDARs), and Ca$^{2+}$ spikes (mediated by voltage-gated Ca$^{2+}$ channels [VGCCs]) (*Stuart and Spruston, 2015*). While Na$^+$ and NMDA spikes can be generated locally in individual thin dendrites of PCs (*Ariav et al., 2003*; *Losonczy and Magee, 2006*; *Makara and Magee, 2013*; *Nevian et al., 2007*; *Polsky et al., 2004*), Ca$^{2+}$ spikes are thought to represent mostly global dendritic events that are responsible for bursting (*Francioni and Harnett, 2021*; *London and Häusser, 2005*; *Magee and Carruth, 1999*; *Stuart and Spruston, 2015*; *Stuyt et al., 2021*; *Williams and Stuart, 1999*).

Studies in hippocampal CA1PCs and neocortical layer 5 PCs (L5PCs) have shown that Ca$^{2+}$ spikes are generated in the main apical trunk efficiently upon widespread synaptic depolarization in distal (tuft) dendrites concomitant with backpropagating action potentials (bAPs), and manifest as an after-depolarization (ADP) producing a characteristic burst of additional APs (also called complex spike burst [CSB]) at the soma (*Harnett et al., 2013*; *Larkum et al., 2009*; *Larkum et al., 1999*; *Takahashi and Magee, 2009*). These results led to a concept that Ca$^{2+}$ spikes can act as an associative dendritic signal that translates a specific input pattern (coincident activation of proximal and distal input pathways) into a burst output (a reliable form of downstream synaptic information transfer; *Lisman, 1997*) and induce synaptic plasticity (*Bittner et al., 2017*; *Takahashi and Magee, 2009*). However, the question emerges: Do Ca$^{2+}$ spikes ubiquitously serve such a canonical role in PCs, or do different PC types express Ca$^{2+}$ spikes with different properties, allowing them to support other input-output transformations and computations?

A particularly interesting PC type to explore these questions are hippocampal CA3 pyramidal cells (CA3PCs). These neurons play a fundamental role in hippocampal associative learning and memory functions (*Kesner, 2013*; *Marr, 1971*; *McNaughton and Morris, 1987*; *Nakazawa et al., 2002*; *Rolls, 2007*), forming a recurrent network governed by input from the dentate gyrus via mossy fibers (MFs) and the entorhinal cortex (EC) (*Witter, 2007*). CA3PCs frequently produce bursts of APs both in vivo and in vitro (*Ding et al., 2020*; *Hablitz and Johnston, 1981*; *Hunt et al., 2018*; *Kowalski et al., 2016*; *Mizuseki et al., 2012*; *Oliva et al., 2016*; *Raus Balind et al., 2019*; *Wong and Prince, 1978*). Investigating CSB generation in CA3PCs, we have recently reported (*Raus Balind et al., 2019*) that CA3PCs are heterogeneous regarding their intrinsic CSB firing propensity, and the required synaptic input patterns for evoking CSBs can be diverse as well: while in a subset of CA3PCs associative inputs drive bursts, in other CA3PCs even inputs restricted to single dendrites can efficiently produce CSBs. However, the dendritic factors underlying this diversity remained to be explored. Little is known about the generation mechanisms and properties of dendritic Ca$^{2+}$ spikes in CA3PCs. Early studies using blind microelectrode recordings observed putative dendritic Ca$^{2+}$ spikes to which bursting was attributed (*Nuñez and Buño, 1992*; *Wong et al., 1979*). Later, direct patch-clamp recordings in CA3PC dendrites revealed and dissected Na$^+$ and NMDA spikes (*Brandalise et al., 2016*; *Kim et al., 2012*; *Makara and Magee, 2013*), but Ca$^{2+}$ spikes have not been specifically examined. Intriguingly, unlike several widely studied PC types (e.g., CA1 or L5) that typically have a long or once bifurcating main apical trunk, the primary apical trunk of CA3PCs bifurcates after a relatively short distance into multiple higher-order intermediate branches. This structure creates separate apical subtrees that may represent independent, parallel integrative units for dendritic processing, allowing interactions between the radially layered inputs: detonator-type MF synapses onto proximal trunks in str. lucidum, recurrent/associative inputs onto str. radiatum dendrites, and long-range EC inputs targeting distal apical branches in str. lacunosum-moleculare. However, whether individual higher-order dendritic families express Ca$^{2+}$ spikes with specific characteristics and function remains unknown.

Here, we performed patch-clamp recordings from higher-order apical dendrites of CA3PCs (alone or simultaneously with their soma) combined with two-photon (2P) Ca$^{2+}$ imaging in the dendritic tree. We report that these dendrites support surprisingly heterogeneous forms of Ca$^{2+}$ spikes that differ from that found in CA1PC dendrites. Besides relatively prolonged ADP-type global Ca$^{2+}$ spikes, a majority of CA3PC dendrites expresses a novel form of fast spike (termed dendritically initiated [DI] spike) that is efficiently triggered without bAPs, is mediated by a combination of fast Ca$^{2+}$ and

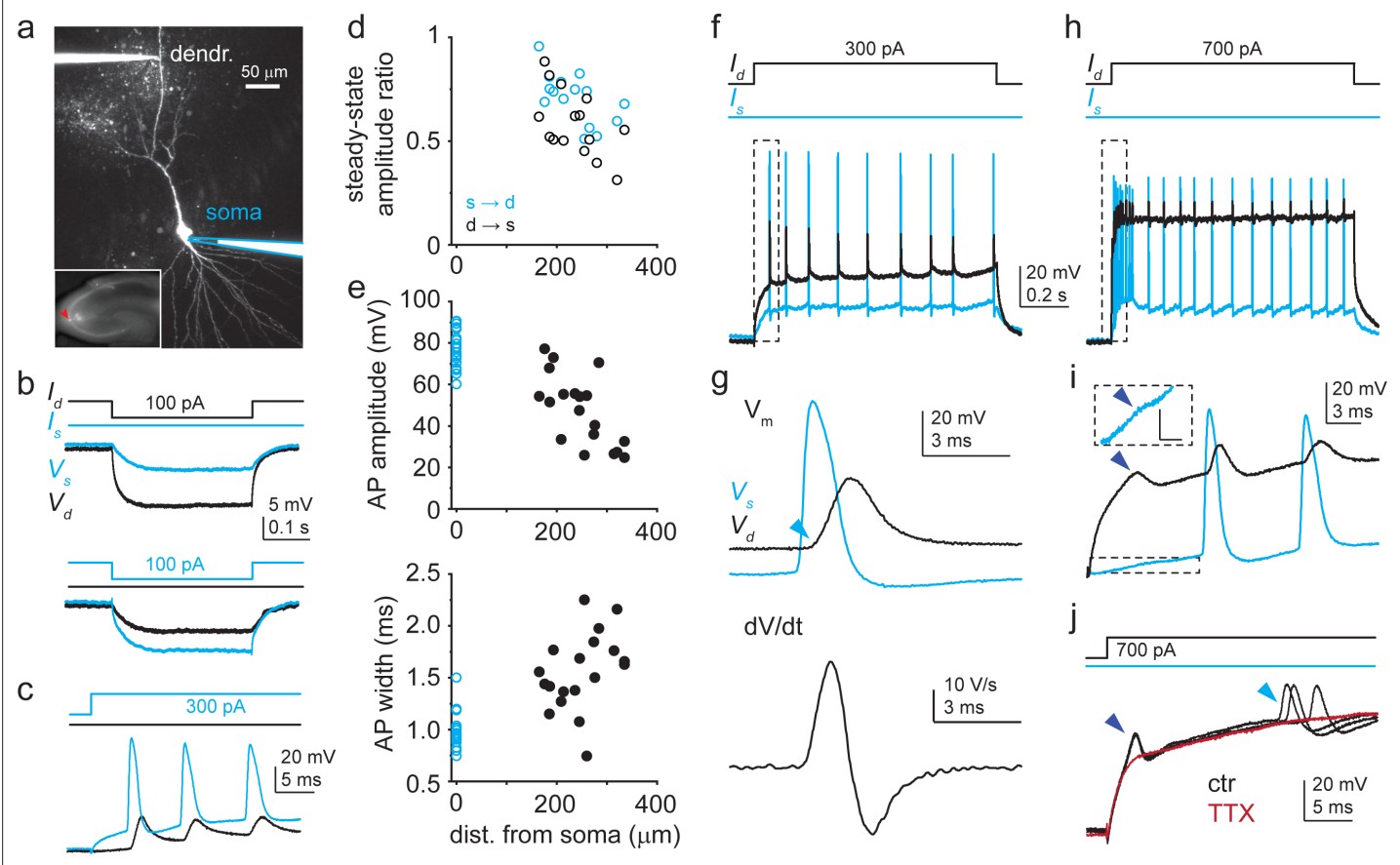

**Figure 1.** Dendritic Na$^+$ spikes in dual recordings of soma and dendrite of CA3 pyramidal cells (CA3PCs). (**a**) Two-photon (2P) collapsed z-stack image of a dually recorded CA3PC (dendritic pipette distance: 255 µm). Somatic pipette outlined in blue. Inset: soma (red arrowhead) located in CA3. (**b**) Somatic (blue) and dendritic (black) voltage responses ($V_s$ and $V_d$) to negative step $I_{inj}$ into the dendrite (top) and the soma (bottom). $I_{inj}$ protocols shown on the top. (**c**) Positive $I_{inj}$ at the soma evokes action potentials (APs) (blue) that backpropagate to the dendrite (black). (**d**) Voltage transfer between soma and dendrite (n = 15 experiments). (**e**) Dendritic distance dependence of amplitude and width at half amplitude of the first backpropagating action potential (bAP) evoked by somatic $I_{inj}$ (n = 19 experiments). (**f**) Voltage responses at the dendrite (black) and at the soma (blue) to a 1-s-long 300 pA dendritic $I_{inj}$ in the cell shown in (**a**). (**g**) First regenerative event in (**f**) (box) is a bAP. Top: voltage; bottom: corresponding dendritic dV/dt. Note the sudden start of the bAP ('kink,' indicated by light blue arrowhead). (**h**) Dendritic and somatic voltage responses to 1-s-long 700 pA dendritic $I_{inj}$ in the cell shown in (**a**). (**i**) Initial part of the voltage response in (**h**) (box) enlarged. A short-latency regenerative dendritic event (dark blue arrowhead) precedes bAPs. Attenuated somatic response is shown enlarged in a dashed box. Scale bar in box: 5 mV, 3 ms. Similar events were observed in three other CA3PCs. (**j**) Short-latency dendritic spikes and APs (black traces; three repetitions) are eliminated by 1 µM TTX (dark red).

The online version of this article includes the following source data and figure supplement(s) for figure 1:

**Source data 1.** Dendritic Na$^+$ spikes in dual recordings of soma and dendrite of CA3 pyramidal cells (CA3PCs).

**Figure supplement 1.** Characteristics of different regenerative dendritic events.

**Figure supplement 1—source data 1.** Characteristics of different regenerative dendritic events.

Na$^+$ currents, and is compartmentalized to the activated dendritic family. Finally, we show that ADP type and DI spikes produce opposing forms of somatic output: bursts versus strictly single APs, and thereby may actively promote different firing modes in morpho-functionally different CA3PCs. The unique properties of DI spikes, such as their fast kinetics and anti-bursting effect, fundamentally differ from the classical associative role of dendritic Ca$^{2+}$ spikes, suggesting that they may enrich the computational repertoire of CA3PCs with novel, cell type- or circuit-specific functions.

## Results

### Identification of regenerative spikes in CA3PCs dendrites

To directly investigate active dendritic properties of CA3PCs, we performed dual soma-dendrite current-clamp recordings combined with two-photon imaging in dendrites (*Figure 1a*). Dendritic recordings were made from higher-order trunks at distances ~165–400 μm from the soma. Subthreshold steady-state voltage signals attenuated more strongly from dendrite to soma than from soma to dendrite (*Figure 1b and d*, n = 15, p=0.004, Wilcoxon test).

Applying 1-s-long depolarizing current injections via either the dendritic or somatic electrode, we readily identified two types of previously described fast VGNC-mediated regenerative voltage responses in the dendrites: bAPs and dendritic Na⁺ spikes (*Kim et al., 2012*). bAPs could be observed in almost all dendrites both by sufficient somatic or dendritic depolarizing stimuli; they followed somatic APs with short latency, their amplitude decreased gradually with distance after the first ~150 μm (*Kim et al., 2012*), and they displayed a sharp initiation profile ('kink,' *Gidon et al., 2020*; *Smith et al., 2013*) and short duration, all characteristic for bAPs (*Figure 1*, *Figure 1—figure supplement 1*). On the other hand, fast dendritic Na⁺ spikes were generated in a subset of experiments (n = 4) by large ($I_{inj}$; ≥ 600 pA) current injections into the dendrite. These events appeared with very short latency (<7 ms) on the initial depolarizing phase of the step and attenuated strongly from dendrite to the soma (*Figure 1h–j*), as previously described (*Kim et al., 2012*). Both of these fast spike types disappeared after the application of 1 μM tetrodotoxin (TTX) in the bath, confirming that they were mediated by VGNCs (*Figure 1j*, n = 12 experiments; local spikes were present in two of these experiments under control conditions; see also later the effect of TTX in single-site dendritic recordings).

In the majority of dual recordings (17 of 21), we also observed slower regenerative voltage responses that we considered to be putative dendritic Ca²⁺ spikes (*Figure 2*). These spikes were elicited with relatively longer latency or at the steady-state depolarized phase, and had highly heterogeneous kinetics (see below). We classified putative Ca²⁺ spikes into two broad groups based on their initiation characteristics. The first, 'classical' group consisted of regenerative ADP forms, that is, responses that followed a bAP with an additional voltage peak (*Figure 2b–d*). ADPs had a wide range of kinetics. In some dendrites, the ADP manifested as a sustained depolarization that triggered additional APs and gradually built up a prolonged, slowly decaying ADP driving a burst of APs ('slow ADP,' >~ 50–100 ms, *Figure 2b*). In other cases, the ADP occurred as a transient regenerative voltage response following 1–3 APs ('fast ADP,' *Figure 2c and d*), which either remained subthreshold or evoked additional AP(s) at the soma.

In addition to 'classical' ADPs, we discovered a second, unconventional form of putative Ca²⁺ spikes. These events were generated without an initiating bAP and were therefore termed dendritically initiated (DI) spikes. The rise of these spikes typically followed a concave, gradually developing profile that could be well distinguished from the sharp 'kink' of bAPs (*Figure 1—figure supplement 1b–e*). Although DI spikes did not require bAPs for their initiation and could be evoked in isolation in the dendrite (*Figure 2e and f*), they could also evoke consecutive bAP (*Figure 2g and h*). The dendritic origin of both ADPs and DI spikes was confirmed by their strong attenuation towards the soma (*Figure 2i and j*, dendrite-soma attenuation ratio: ADPs: 5.06 ± 1.28, n = 7; DI spikes: 6.97 ± 1.09, n = 5). ADPs and DI spikes co-occurred in some of the recordings (*Figure 2k*).

Simultaneously with electrophysiology, we measured dendritic Ca²⁺ signals (using OGB-1 or OGB-6F, see Materials and methods) at locations > 89 μm (mean ± SEM.: 234 ± 20 μm) distal from the dendritic pipette. All forms of the above-identified putative Ca²⁺ spikes (but not local Na⁺ spikes, *Figure 1—figure supplement 1a*) were accompanied by large distal dendritic Ca²⁺ signals that coincided with the onset of the voltage response (*Figure 2b, c, e and g*), suggesting that the voltage signals propagated actively and involved regenerative Ca²⁺ influx.

### Characterization of dendritic Ca²⁺ spikes

Equipped with the fingerprint characteristics to identify putative Ca²⁺ spikes (see also *Figure 1—figure supplement 1*), we further characterized the properties of these spikes using $I_{inj}$ in a larger set of single-site dendritic recordings (n = 69, at 159–450 μm from soma, mean ± SEM: 272 ± 8 μm). Replicating the findings of dual recordings, 1-s-long dendritic $I_{inj}$ steps elicited different types of putative Ca²⁺ spikes, that is, ADPs and/or DI spikes in the majority (~85%) of individual dendrites, in addition to bAPs and dendritic Na⁺ spikes (*Figure 3a–d and f*, *Figure 1—figure supplement 1*). Putative Ca²⁺

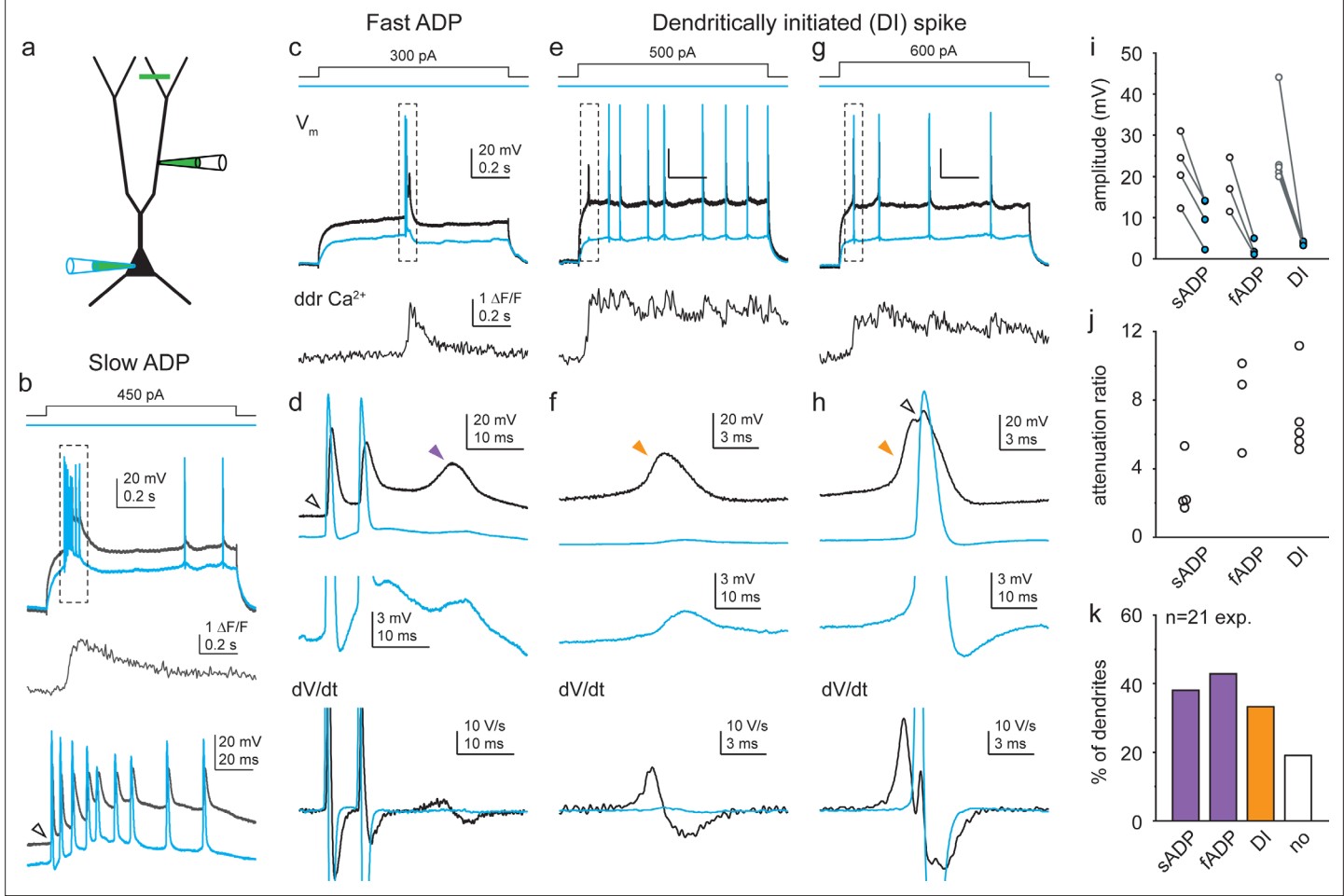

**Figure 2.** Diverse dendritic Ca²⁺ spike forms in soma-dendrite dual recordings from CA3 pyramidal cells (CA3PCs). (**a**) Schematic of experiments. Green line indicates typical Ca²⁺ imaging site distal from the dendritic patch. (**b**) Representative recording of a slow afterdepolarization (ADP). Dendritic (black) and somatic (blue) voltage response pair to dendritic $I_{inj}$ (top), and corresponding distal dendritic Ca²⁺ signal (middle). Dashed box on top indicates the event that is enlarged on the bottom. Note the prolonged sustained ADP building a slow depolarization that is larger in the dendrite. Open arrowhead denotes kink of the initiating backpropagating action potential (bAP). (**c**) Representative recording of a fast ADP. (**d**) Event in dashed box in (**c**) is shown enlarged. Top: dendritic (black) and somatic (blue) voltage pair. Middle: somatic trace magnified. Bottom: corresponding dV/dt traces (action potentials [APs] truncated). Open arrowhead denotes 'kink'; purple arrowhead points to the fast ADP. (**e–g**) Representative recordings of dendritically initiated (DI) spikes, either isolated in the dendrite (**e, f**) or evoking a consecutive AP (**g, h**). Panels as in (**c, d**). Open arrowhead denotes AP with 'kink'; orange arrowheads indicate DI spikes. Traces in (**b, c, e, g**) are from different cells. Note the different time scales of various spike forms. (**i**) Ca²⁺ spike amplitudes in dendrite (open circle) and soma (blue circle) in individual recordings. See Materials and methods for details. (**j**) Calculated attenuation ($Ampl_{ddr}/Ampl_{soma}$) in individual recordings. (**k**) Propensity of different Ca²⁺ spike forms (in total n = 21 dendrites with dual recordings).

The online version of this article includes the following figure supplement(s) for figure 2:

**Source data 1.** Diverse dendritic Ca²⁺ spike forms in soma-dendrite dual recordings from CA3 pyramidal cells (CA3PCs).

spikes were often generated repetitively, and in some cells their kinetics were variable even across repetitions (*Figure 3—figure supplement 1*). The spikes were unaffected by blockade of AMPA and NMDA receptors, confirming that they were not related to excitatory synaptic activity (*Figure 3—figure supplement 2*).

In a set of dendrites, we systematically determined the propensity of the two main Ca²⁺ spike types evoked in the 300–600 pA $I_{inj}$ range, and we found that ADPs and DI spikes were expressed in 56 and 53% of the investigated dendrites, respectively (*Figure 3f*, n = 70 cells, dual- and single-site experiments pooled), and in 24% they both were present. The dominant Ca²⁺ spike type often depended on the $I_{inj}$ level, with ADPs evoked at smaller and DI spikes evoked at higher $I_{inj}$ (*Figure 3c and f*). Furthermore, in some dendrites different Ca²⁺ spike types occurred intermingled within the same depolarizing trace (*Figure 3d*). As in dual recordings, both ADPs and DI spikes were associated

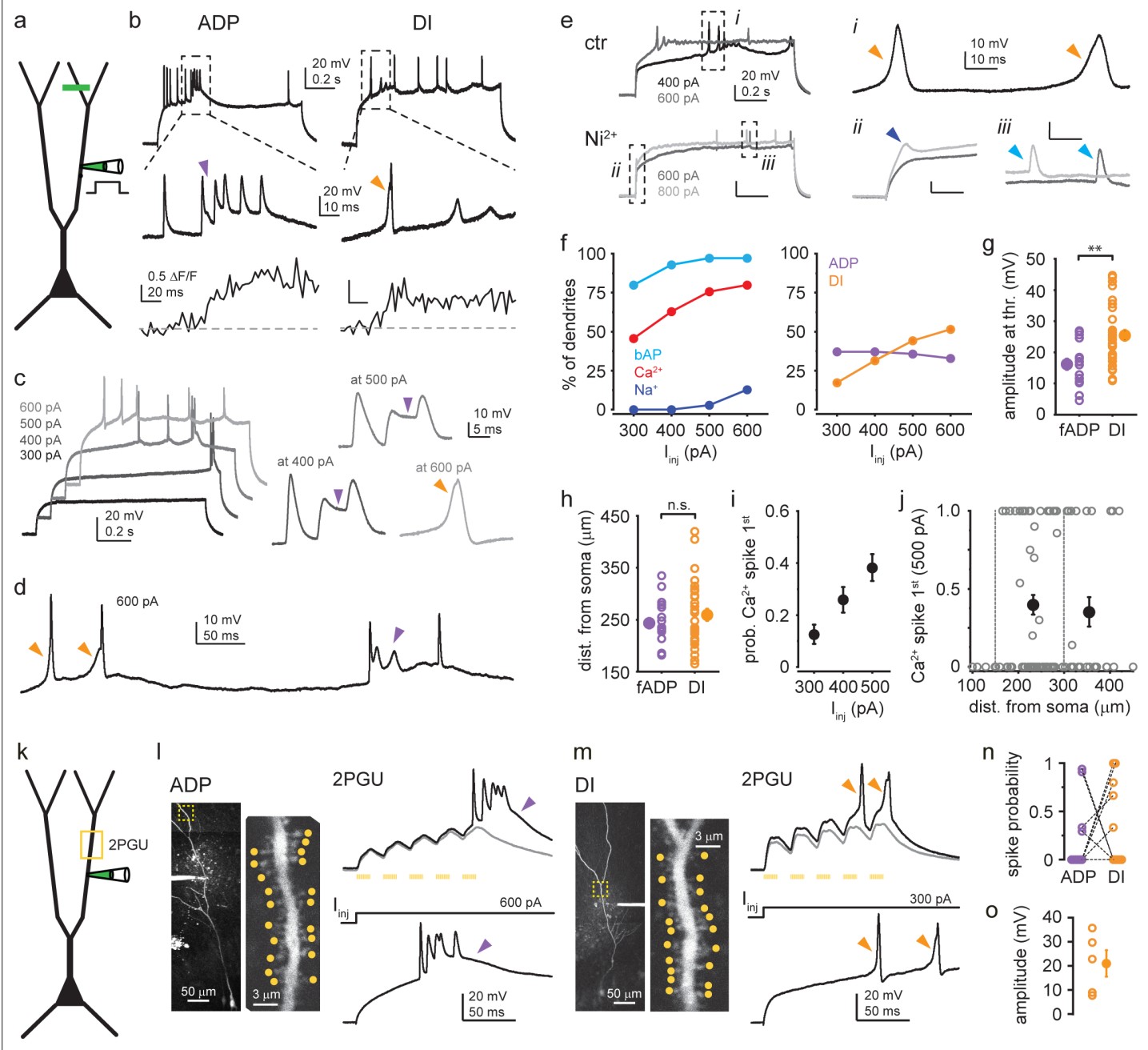

**Figure 3.** Characterization of dendritic Ca²⁺ spike types. (**a**) Schematic of dendrite-only experiments. (**b**) Representative dendritic voltage and Ca²⁺ recording of afterdepolarization (ADP) (left) and dendritically initiated (DI) spike (right) from two different cells. (**c**) Example responses of a dendrite to increasing $I_{inj}$. Right: Ca²⁺ spikes expressed on different traces magnified. Purple arrowheads: ADP; orange arrowhead: DI spike. (**d**) Single trace in response to 600 pA $I_{inj}$, containing heterogeneous Ca²⁺ spike forms. (**e**) Representative traces under control conditions (top, two different $I_{inj}$) and after bath application of 200 μM Ni²⁺ (bottom). Dashed boxes are enlarged on the right to show DI spikes (i, orange arrowhead), dendritic Na⁺ spike (ii, dark blue arrowhead), and backpropagating action potentials (bAPs) (iii, light blue arrowhead). Note that Na⁺ spikes were resistant to Ni²⁺ (see also *Figure 1—figure supplement 1i*). (**f–j**) Summary of Ca²⁺ spike properties (dendrite-only and dual recordings pooled). (**f**) Left: percent of dendrites expressing bAPs, Ca²⁺ spikes (ADPs and DI spikes included), and Na⁺ spikes to 300–600 pA $I_{inj}$ (n = 70 dendrites). Right: percent of dendrites expressing different types of Ca²⁺ spikes (ADPs and DI Ca²⁺ spikes) to 300–600 pA $I_{inj}$ (n = 70 dendrites). (**g**) Amplitude of fast ADPs and DI spikes at threshold $I_{inj}$. Open circles: mean amplitude in individual dendrites; filled symbol: mean ± SEM of experiments. (**h**) Dendritic distance of pipette from the soma in the experiments in (**g**). (**i**) Mean probability (range: 0–1) that Ca²⁺ spike was the first regenerative event evoked by 300, 400, or 500 pA $I_{inj}$ (n = 70, 75, 82 dendrites). (**j**) Probability (range: 0–1) that Ca²⁺ spike was the first regenerative event evoked by 500 pA $I_{inj}$ as a function of pipette distance from soma. Additional proximal recordings are also shown. Open gray circles: individual dendrites; filled black symbols: mean ± SEM of measurements in 151–300 μm (n = 56) and 301–450 μm (n = 26) distance range from soma. (**k**) Schematic of 2P glutamate uncaging (2PGU) experiments. (**l**) Left: z-stack of

*Figure 3 continued on next page*

*Figure 3 continued*

a CA3PC, and single scan of the dendritic segment (marked by yellow dashed box) indicating the 20 synapses stimulated by 2PGU (yellow dots). Right, top: example responses to 2PGU (20 spines stimulated quasi-synchronously 5× at 40 Hz). Gray: subthreshold; black: suprathreshold response. Bottom: same dendrite responding to $I_{inj}$ via the pipette. (**m**) Same as (**l**) for a dendrite with DI spike. Note the similarities in spike types by 2PGU and $I_{inj}$ in (**l–m**). (**n**) Relative probability of evoking ADPs and DI spikes in dendrites (number of traces displaying the respective d-spike divided by the total number of suprathreshold traces). Dashed lines connect data from the same dendrites. (**o**) Amplitude of DI spikes evoked by 2PGU.

The online version of this article includes the following source data and figure supplement(s) for figure 3:

**Source data 1.** Characterization of dendritic $Ca^{2+}$ spike types.

**Figure supplement 1.** Variability of $Ca^{2+}$ spike phenotypes across and within CA3 pyramidal cells (CA3PCs).

**Figure supplement 2.** Dendritic $Ca^{2+}$ spike properties do not depend on excitatory synaptic activity.

**Figure supplement 2—source data 1.** Dendritic $Ca^{2+}$ spike properties do not depend on excitatory synaptic activity.

**Figure supplement 3.** Dendritic $Ca^{2+}$ spikes in CA1 pyramidal cells (CA1PCs) are afterdepolarization (ADP) type.

**Figure supplement 3—source data 1.** Dendritic $Ca^{2+}$ spikes in CA1 pyramidal cell (CA1PCs) are afterdepolarization (ADP) type.

with large $Ca^{2+}$ signals measured in dendritic segments > 80 μm (mean ± SEM: 215 ± 9 μm) distal to the patch pipette (*Figure 3b*). Bath application of 200 μM $Ni^{2+}$ eliminated all types of putative $Ca^{2+}$ spikes, whereas bAPs and dendritic $Na^+$ spikes were spared (*Figure 3e, Figure 1—figure supplement 1i*). These results confirmed that VGCCs played a fundamental role in mediating ADPs and DI spikes. Interestingly, the amplitude of DI spikes, measured at threshold $I_{inj}$ (see Materials and methods for amplitude measurement criteria), was larger (25.4 ± 1.7 mV, n = 30) than that of fast ADPs (16.2 ± 1.8 mV, n = 18; Mann–Whitney test p=0.003, *Figure 3g*). This difference was not explained by different dendritic distance of the recordings from the soma (DI spikes: 259 ± 12 μm, ADPs: 243 ± 10 μm, Mann–Whitney test p=0.601, *Figure 3h*), and the amplitude of neither spike type correlated with distance (Spearman correlation: ADP: $R = −0.260$, p=0.296, n = 18; DI: $R = −0.297$, p=0.110, n = 30), suggesting a difference in the spike generation mechanism rather than simple distance-dependent variation of spike properties.

We next addressed the question of whether the soma or the dendrite is more likely to first generate a regenerative spike upon dendritic depolarization. When the patch pipette was positioned at relatively proximal dendritic locations (<~150 μm from soma), somatic APs were always evoked first by dendritic depolarization. However, at more distal dendritic locations DI $Ca^{2+}$ spikes were often evoked before bAPs, and this propensity depended on the strength of $I_{inj}$ (*Figure 3i*, ratio at 300 pA: 0.13 ± 0.04, n = 70; 400 pA: 0.26 ± 0.05, n = 75; 500 pA: 0.38 ± 0.05, n = 82; p=0.001, Kruskal–Wallis test) but was independent from the distance from soma (*Figure 3j*, Spearman correlation $R = −0.031$, p=0.778, n = 82). Thus, DI spikes can be evoked efficiently by local depolarization of medial-distal apical trunks of CA3PCs without any preceding somatic activity.

To test whether the distinct forms of $Ca^{2+}$ spikes observed by $I_{inj}$ could be also evoked by more physiological forms of stimulation, we next performed experiments employing synaptic stimulation on the trunk using 2P glutamate uncaging (2PGU; *Figure 3k and l*). We patched dendrites (n = 10 experiments, 230 ± 17 μm from soma) and stimulated 20 clustered spines located along the same dendrite at 171–441 μm distance from the soma quasi-simultaneously 5× at 40 Hz by adjusting the laser power to produce moderately suprathreshold stimulation (bAP or dendritic spike evoked by any of the last three of the five stimuli). To avoid activation of confounding slow NMDA spikes, these experiments were performed in the presence of an NMDAR blocker in the bath (D-AP5, 50 μM). Synaptic stimulation was able to elicit characteristic and well-distinguishable ADPs and DI spikes (*Figure 3n*) either separately (ADP only: 3/10 dendrites, *Figure 3l*; DI spike only: 4/10 dendrites, *Figure 3m*) or in combination (1/10 dendrites). In the two dendrites with no d-spikes by moderate stimulation, stronger stimuli could activate ADPs (data not shown). The $Ca^{2+}$ spike profile of the dendrite evoked by uncaging was similar to that seen with $I_{inj}$ via the pipette (*Figure 3l and m*), and DI spikes had amplitudes comparable to that typically evoked by $I_{inj}$ (*Figure 3o*). These results confirm that the $Ca^{2+}$ spike form is an inherent property of the dendrite, and separate $Ca^{2+}$ spike modes can be elicited by a wide range of stimuli that reach sufficiently strong local depolarization.

The surprisingly variable properties of putative dendritic $Ca^{2+}$ spikes found in CA3PC dendrites prompted us to compare these spikes to the well-characterized $Ca^{2+}$ spikes of CA1PCs (*Golding et al., 1999; Magee and Carruth, 1999; Takahashi and Magee, 2009*). Therefore, we performed

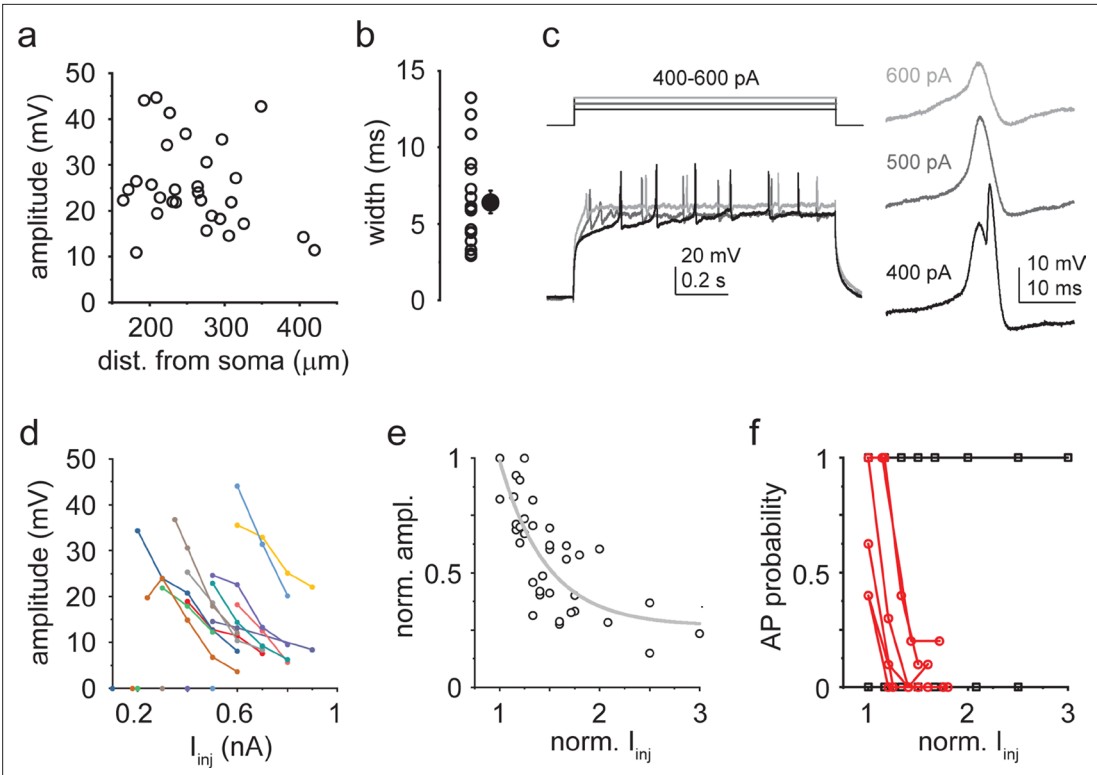

**Figure 4.** Properties of dendritically initiated (DI) Ca²⁺ spikes. (**a**) Spike amplitude as a function of pipette distance from soma (circles: individual dendrites, n = 30). (**b**) Width at half amplitude of DI spikes. Open circles: individual dendrites; filled symbol: mean ± SEM (n = 18). (**c**) Representative responses of a dendrite to 400–600 pA $I_{inj}$. Right: first regenerative events enlarged. (**d**) $I_{inj}$–amplitude response curves in 12 dendrites with DI spikes. (**e**) Normalized $I_{inj}$–amplitude relationship established from the data in (**d**). Gray line: exponential decay function. (**f**) Probability of action potential (AP) firing directly elicited by DI spikes at various $I_{inj}$ levels. In 6 out of 12 dendrites, larger $I_{inj}$ evoked DI spikes with progressively lower AP probability (red). In the other six dendrites, AP probability remained 0 or 1.

The online version of this article includes the following figure supplement(s) for figure 4:

**Source data 1.** Properties of DI Ca²⁺ spikes.

similar experiments in the apical trunks of CA1PCs, at 227–457 μm distance from soma (n = 12, **Figure 3—figure supplement 3**). In most CA1PCs, dendritic depolarization by 300–600 pA $I_{inj}$ evoked VGCC-mediated ADPs (**Figure 3—figure supplement 3a–c, g**), but the required $I_{inj}$ to trigger the spikes was higher than that in CA3PCs (**Figure 3—figure supplement 3d**). Furthermore, DI spikes were not observed in CA1PC trunks (**Figure 3—figure supplement 3e**), and accordingly, bAPs were always evoked before Ca²⁺ spikes (**Figure 3—figure supplement 3f**).

Thus, active properties of CA3PC trunk dendrites fundamentally differ from those of CA1PCs, indicating that the repertoire and roles of dendritic computation can vary in different types of rat PCs. Curiously, however, some features of CA3PC dendritic Ca²⁺ spikes resembled those recently described in L2/3PCs in cortical slices removed from human patients (hL2/3PCs; **Gidon et al., 2020**). Specifically, the DI spikes we observed in CA3PCs shared several properties with a novel type of Ca²⁺ spikes in hL2/3 PCs: they were characterized by bAP-independent initiation, fast rise and large amplitude (**Figure 4a and b**), which inversely scaled with increasing $I_{inj}$ (**Figure 4c–e**). In some dendrites (6 out of 12 tested), the depolarization level-dependent reduction in amplitude created a window of dendritic stimulus strength where APs could be specifically triggered by these dendritic events (**Figure 4f**), similarly to that described in hL2/3PCs (**Gidon et al., 2020**). A systematic comparison of the parameters of DI spikes in CA3PCs to those reported for hL2/3PCs revealed remarkable qualitative similarities with moderate quantitative differences (CA3PC: amplitude at threshold $I_{inj}$: 25.43 ± 1.74 mV, n = 30; width at half amplitude: 6.43 ± 0.75 ms, n = 18; exponential decay constant [tau] of normalized amplitude vs. $I_{inj}$: 0.47; hL2/3PC parameters from **Gidon et al., 2020**: amplitude: 43.8 ±

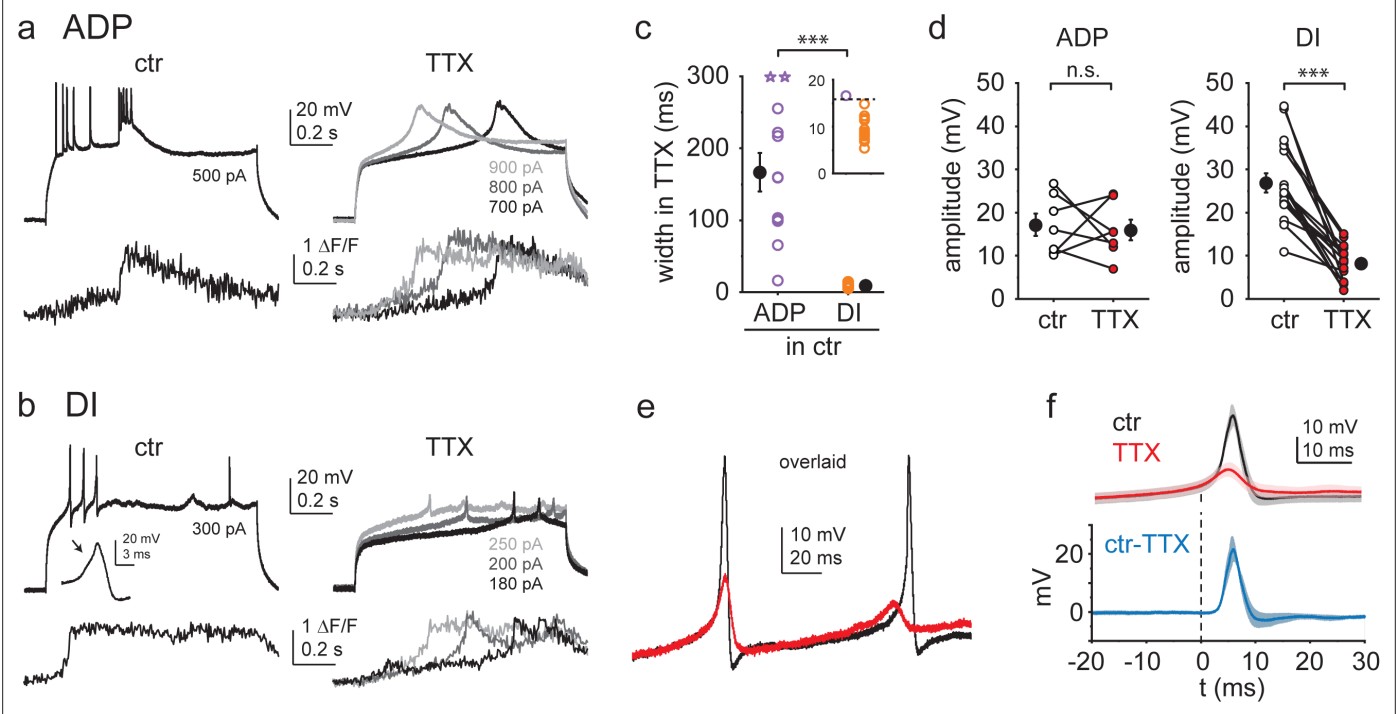

**Figure 5.** Ca²⁺ spikes with different kinetics are mediated by different ionic mechanisms. (**a, b**) Representative experiments showing the effect of 1 µM TTX on afterdepolarizations (ADPs) (**a**) and dendritically initiated (DI) spikes (**b**). $V_m$ (top) and distal dendrite Ca²⁺ signals (bottom) under control conditions (left) and after application of TTX at various $I_{inj}$ levels. (**c**) Width of TTX-resistant Ca²⁺ spikes in cells with ADP only (purple, n = 12) and cells expressing DI spikes (orange, n = 13). Open circles: individual dendrites (open stars: dendrites where width was maximized as 300 ms because $V_m$ did not return to half amplitude within the duration of the $I_{inj}$ step); filled black symbols: mean ± SEM. Inset shows the 0–20 ms width range magnified. (**d**) Comparison of spike amplitude before and after TTX application in dendrites with ADPs only (left, n = 7) and dendrites with DI spikes (right, n = 17). (**e**) Overlaid voltage traces from a dendrite in control (black) and after TTX application (red). (**f**) Summary of the impact of TTX on DI spike kinetics. Top: $V_m$ traces before (black, ctr) and after (red) TTX application (mean ± SEM of six experiments), aligned to 1 V/s. Bottom: result of the subtraction of ctr and TTX traces. The fast TTX-sensitive component follows the initial slow depolarization.

The online version of this article includes the following source data and figure supplement(s) for figure 5:

**Source data 1.** Ca²⁺ spikes with different kinetics are mediated by different ionic mechanisms.

**Figure supplement 1.** Additional properties of dendritic Ca²⁺ spikes.

**Figure supplement 1—source data 1.** Additional properties of dendritic Ca²⁺ spikes.

13.8 mV; width: 4.4 ± 1.4 ms; tau: 0.39). We conclude that DI spikes are expressed in specific neuron types of various species including humans and rats.

## Ion channel mechanisms

To study the properties of CA3PC Ca²⁺ spikes in isolation, we bath applied the VGNC inhibitor TTX (1 µM). As expected, TTX completely eliminated bAPs and Na⁺ spikes (***Figure 1—figure supplement 1h and i***, see also ***Figure 1***), whereas putative Ca²⁺ spikes, associated with large dendritic Ca²⁺ signals, persisted. The kinetics of the regenerative spikes remaining in TTX varied in a wide range, but mirrored the behavior in control conditions. That is, in dendrites that under control conditions expressed only ADPs (fast or slow) but no DI spikes, TTX-resistant Ca²⁺ spikes were typically slow (***Figure 5a and c***, width: 166.6 ± 26.8 ms, n = 12), whereas those dendrites that fired DI spikes (with or without ADPs) under control conditions displayed dominantly fast, transient, TTX-resistant Ca²⁺ spikes (***Figure 5b and c***, width: 9.3 ± 0.7 ms, n = 13, Mann–Whitney test p<0.001), although a smaller slow component was often also present. In some cases (mostly in cells expressing both ADPs and DI spikes in artificial cerebrospinal fluid (ACSF)), slow and repetitive fast components were mixed (***Figure 5—figure supplement 1a***). The duration of TTX-resistant Ca²⁺ spikes in CA3PCs clearly separated from those of CA1PC dendrites under similar conditions (width: 39.4 ± 5.1 ms, n = 6, ***Figure 3—figure supplement 3h and i***), further confirming cell-type-specific differences in Ca²⁺ spike properties. Finally, in ~25% of

dendrites, no clear regenerative voltage responses could be evoked after TTX application (*Figure 5— figure supplement 1b*). These results, together with the elimination of all types of $Ca^{2+}$ spikes by 200 µM $Ni^{2+}$ (*Figure 3e*), confirm a fundamental role of VGCCs in generating a wide kinetic range of $Ca^{2+}$ spikes in CA3PCs.

To our surprise, we noticed that the amplitude (and dV/dt) of TTX-resistant fast spikes was consistently smaller than that of the DI spikes in ACSF (in ACSF: 26.9 ± 2.2 mV, in TTX: 8.2 ± 1.0 mV, n = 17, Wilcoxon test, p<0.001), whereas the amplitudes of ADPs and TTX-resistant slow spikes were not systematically different (17.2 ± 2.6 mV before and 15.9 ± 2.4 mV after TTX application, n = 7, p=0.735, Wilcoxon test, *Figure 5d*). Therefore, we examined in more detail the possibility that VGNCs contribute in some form selectively to DI spikes. First, we determined which parameters of the spike were most affected by TTX. Our analysis showed that the reduction of spike amplitude was largely caused by a drop in peak voltage (from –0.8 ± 2.2 mV to –15.6 ± 1.3 mV, n = 17, p<0.001, Wilcoxon test, *Figure 5—figure supplement 1d*) rather than the modest increase in spike threshold (from –27.2 ± 1.0 mV to –23.7 ± 1.0 mV, n = 17, p<0.01, Wilcoxon test, *Figure 5—figure supplement 1e and f*). Furthermore, overlaying individual traces (*Figure 5e*) or aligning averaged voltage responses in

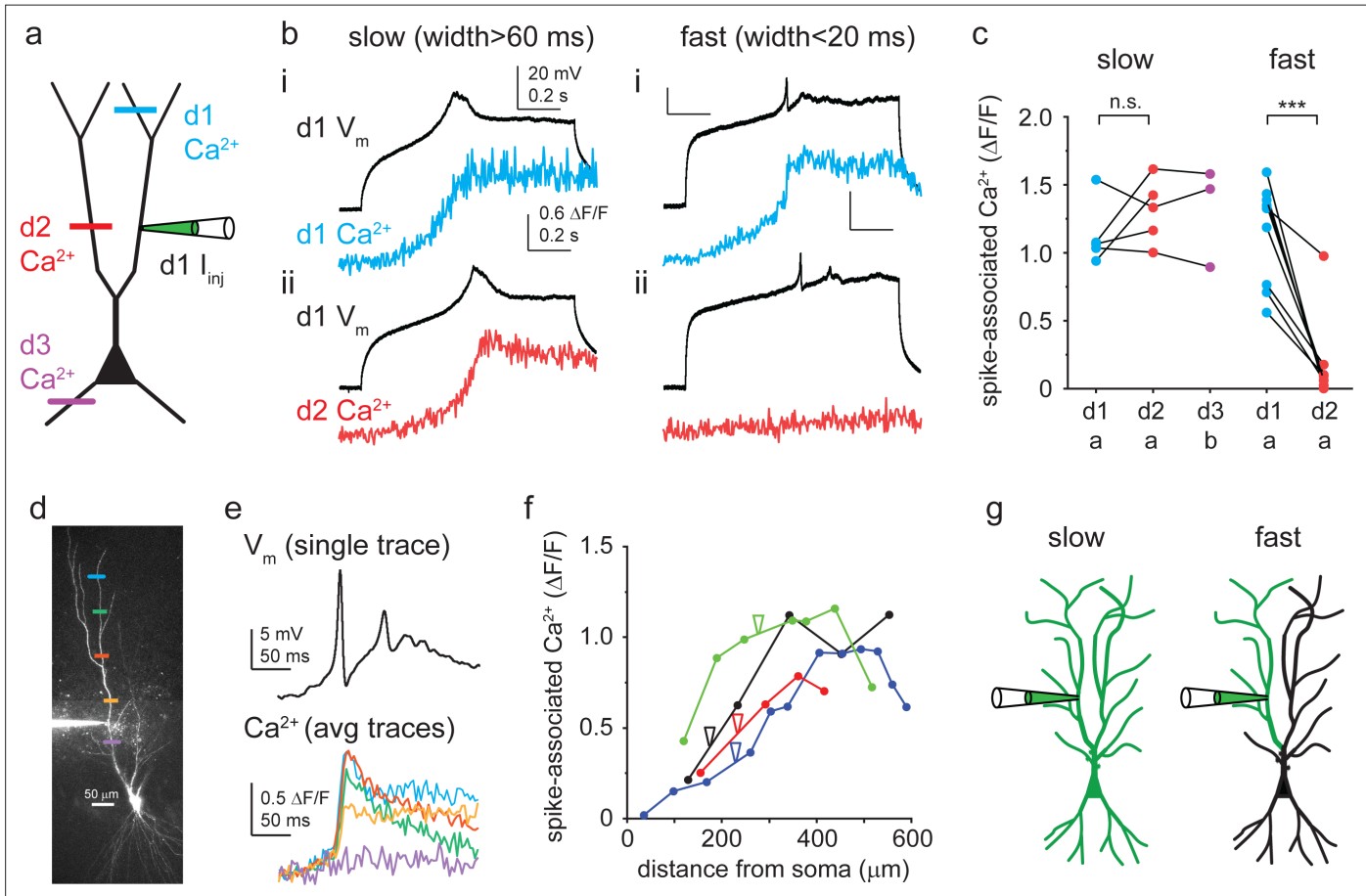

**Figure 6.** Propagation of different $Ca^{2+}$ spike types. (**a**) Schematic of experimental strategy. Dendritic $Ca^{2+}$ signals were measured as a proxy to assess $Ca^{2+}$ spike propagation in different regions of the arbor. (**b**) Representative TTX-resistant slow (left) and fast (right) $Ca^{2+}$ spikes. Spike-evoked $Ca^{2+}$ signals (OGB-1) measured in the patched (i) and in a different (ii) apical dendritic subtree. (**c**) Summary of spike-evoked $Ca^{2+}$ signal amplitudes at different dendritic tree parts. Note that large spike-evoked $Ca^{2+}$ signals likely saturate OGB-1. (**d**) Two-photon (2P) stack of a CA3PC, with $Ca^{2+}$ measurement sites indicated by colored lines. (**e**) Fast $Ca^{2+}$ spike-evoked $Ca^{2+}$ signals (measured with OGB-6F) at the locations indicated in panel (**d**). $Ca^{2+}$ signals were aligned to spike onset (one voltage trace shown on top). (**f**) Distance dependence of fast $Ca^{2+}$ spike-associated $Ca^{2+}$ signals in four experiments. Note the drop of $Ca^{2+}$ signals from the pipette towards the soma. (**g**) Concept of compartmentalization rules. Slow $Ca^{2+}$ spikes are global events, whereas fast $Ca^{2+}$ spikes are restricted to apical dendritic subtrees.

The online version of this article includes the following figure supplement(s) for figure 6:

**Source data 1.** Propagation of different $Ca^{2+}$ spike types.

control and TTX to a specific dV/dt value (1 V/s; *Figure 5f*) revealed that TTX did not affect the initial slow, concave rise of the spike, but exclusively reduced the subsequent, fast-rising peak component. As an independent confirmation of the involvement of VGNCs in the generation of DI spikes, replacement of a majority of extracellular $Na^+$ ions in the ACSF with the large cation $NMDG^+$ (which permeates less through VGNCs) produced a similar effect to that of TTX (*Figure 5—figure supplement 1c and g*). Altogether, these results suggest that DI spikes can be mediated by a hybrid mechanism, whereby initiation of a fast $Ca^{2+}$ spike can recruit regenerative activation of VGNCs that further amplify and sharpen the voltage response, with the two components blending smoothly into a rapid and transient combined dendritic spike.

## Different $Ca^{2+}$ spike types obey distinct compartmentalization rules

The distinct characteristics of fast and slow $Ca^{2+}$ spike components raise the question whether their propagation and compartmentalization properties are also different. To address this, we recorded $Ca^{2+}$ spike-associated dendritic $Ca^{2+}$ signals (in TTX) both distally within the same dendritic family (250 ± 28 µm distal from the patch pipette, n = 14 cells; depicted as d1 in *Figure 6a*) and in another dendrite branching off more proximally, typically from a different low-order trunk segment (d2; dendritic distance from pipette: 294 ± 23 µm, n = 14, *Figure 6a*). We first used the high-affinity dye OGB-1 to be able to detect even small increases in $Ca^{2+}$ as a reporter of spike propagation within and across dendritic compartments. We found a strong difference between fast and slow spikes in their propagation properties (two-way repeated measures ANOVA, p<0.001 for kinetic group, p=0.005 for location, p<0.001 for interaction). Slow $Ca^{2+}$ spikes (width >60 ms) were accompanied by large (likely dye-saturating) $Ca^{2+}$ signals both in d1 and d2 (*Figure 6b and c*, n = 5 cells, p=0.781, Tukey's post hoc test), and even in basal dendrites (measured in n = 3 cells, *Figure 6d*), indicating efficient propagation across major bifurcation points or even the entire dendritic tree. In contrast, fast $Ca^{2+}$ spikes (width <20 ms) were restricted to the dendritic family connected to the patched trunk (*Figure 6b and c*; n = 9 cells, p<0.001, Tukey's post hoc test). Finally, to assess the local propagation capacity of fast $Ca^{2+}$ spikes in more detail, we mapped fast $Ca^{2+}$ spike-associated $Ca^{2+}$ signals in higher spatial resolution proximally and distally from the patch pipette (using the low-affinity $Ca^{2+}$ dye OGB-6F to reduce complications from dye saturation). These experiments revealed relatively uniform, large $Ca^{2+}$ signals towards distal dendritic locations, but a strong drop of $Ca^{2+}$ signals within ~160 µm from the pipette in the proximal direction, so that fast spikes evoked in a higher-order trunk did not propagate closer than ~100 µm to the soma (*Figure 6d–f*, n = 4 experiments).

These results indicate different generation and propagation properties of different $Ca^{2+}$ spike types (*Figure 6g*). Slow $Ca^{2+}$ spikes are apparently global events that engage virtually the entire dendritic tree. In contrast, fast $Ca^{2+}$ spikes are generated within the dendritic family of the stimulated trunk, and although they propagate well distally, they cannot efficiently invade the proximal main or sibling trunk segments, creating a mesoscale compartmentalization level represented by separate apical dendritic subtrees.

## Morphological correlates of distinct active dendritic properties

The dendritic structure of CA3PCs is highly diverse, and their electrophysiological properties were proposed to be related to their topographic position within the CA3 area and dendritic morphology (*Bilkey and Schwartzkroin, 1990*; *Ding et al., 2020*; *Hunt et al., 2018*; *Raus Balind et al., 2019*; *Sun et al., 2017*). A recent study even suggested the existence of a sparse class of CA3PCs with bursting phenotype that lack thorny excrescences (TEs) and input from mossy fibers ('athorny' cells, *Hunt et al., 2018*). Therefore, we examined the relationship between $Ca^{2+}$ spike phenotype and morphological properties of CA3PCs in our dataset. In almost all of our fluorescently labeled cells, we unambiguously confirmed the presence of TEs. The estimated coverage of the proximal trunk(s) by TEs varied widely (*Figure 7a and b*) and was inversely correlated with the length of first-order apical trunk(s) (*Figure 7c*, Spearman correlation R = −0.495, p<0.001, n = 89), confirming previous studies (*Fitch et al., 1989*) and suggesting a gradient of morpho-functional properties of regular CA3PCs. We found that the dendritic $Ca^{2+}$ spike profile was correlated with these basic anatomical features of CA3PCs. First, CA3PCs expressing DI spikes in the recorded dendrite (with or without ADPs) had on average twice as high TE coverage (*Figure 7d*, ADP-only: 40 ± 5 µm, n = 19; DI: 84 ± 8 µm, n = 35, Mann–Whitney test: p<0.001) than ADP-only cells. Second, cells with DI spikes had shorter primary

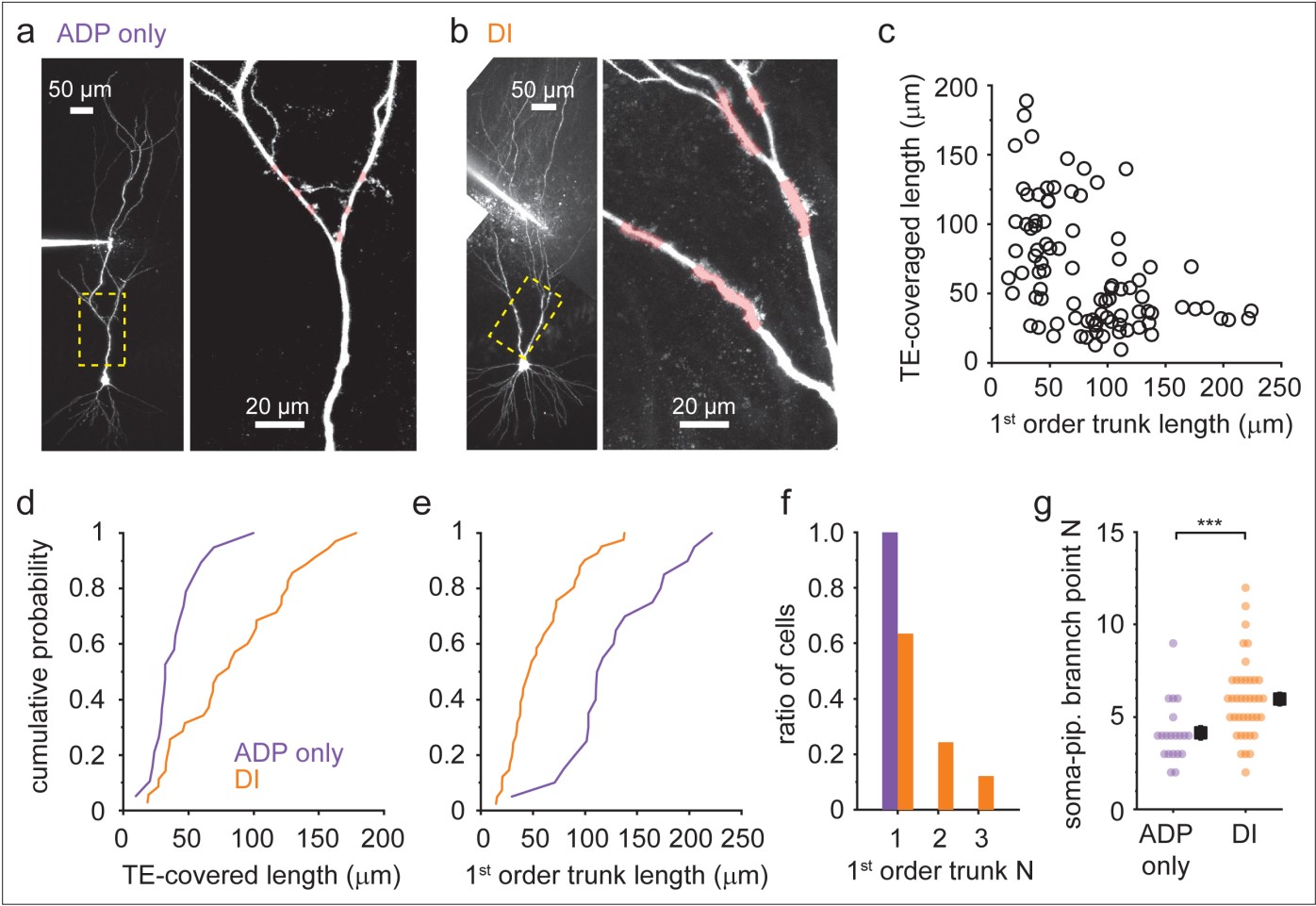

**Figure 7.** Dendritic Ca$^{2+}$ spike phenotype correlates with morphological traits (**a, b**). Representative two-photon (2P) stack of two CA3 pyramidal cells (CA3PCs) with afterdepolarization (ADP)-only (**a**) or dendritically initiated (DI) (**b**) Ca$^{2+}$ spike phenotypes (with ≤600 or 700 pA I$_{inj}$). Yellow dashed boxes are enlarged on the right panels. Trunk segments with thorny excrescences (TEs) are indicated by pink shading. (**c**) Relationship between first-order apical trunk length and total TE-covered dendrite length among CA3PCs (n = 89). For cells with multiple primary trunks, the mean trunk length is shown. (**d, e**) Cumulative probabilities of total TE-covered dendrite length (**d**, n = 19 ADP-only, n = 35 DI) and first-order apical trunk length (**e**, n = 20 ADP-only, n = 41 DI) for cells with dendritically recorded ADP-only and DI Ca$^{2+}$ spike types. (**f**) Number of first-order apical trunks for the two electrophysiological groups (n = 20 ADP-only, n = 41 DI). (**g**) Number of branch points between the patch pipette and the soma in the two electrophysiological groups.

The online version of this article includes the following figure supplement(s) for figure 7:

**Source data 1.** Dendritic Ca$^{2+}$ spike phenotype correlates with morphological traits.

trunks (**Figure 7e**, ADP-only: 128 ± 11 μm, n = 20; DI: 57 ± 5 μm, n = 41, Mann–Whitney test: p=0.002) and more often had multiple first-order trunks (**Figure 7f**, ADP-only: 1 ± 0, n = 20; DI: 1.49 ± 0.11, n = 41, Mann–Whitney test: p<0.001) than ADP-only cells. Interestingly, although the distance of the dendritic patch pipette from the soma was similar in the two electrophysiological groups (ADP-only: 276 ± 15 μm, n = 20; DI: 260 ± 10 μm, n = 41, Mann–Whitney test: p=0.282, see also **Figure 3h**), the dendritic path between the pipette and soma contained more branch points in the case of dendrites with DI spikes (**Figure 7g**, ADP-only: 4.15 ± 0.36, n = 20; DI: 5.97 ± 0.35, n = 39, Mann–Whitney test: p<0.001). This may be consistent with the idea that dendrites with more branch points are electrically more isolated from the soma (**Vetter et al., 2001**). In summary, the morphological diversity of CA3PCs corresponds to distinct dendritic excitability phenotypes.

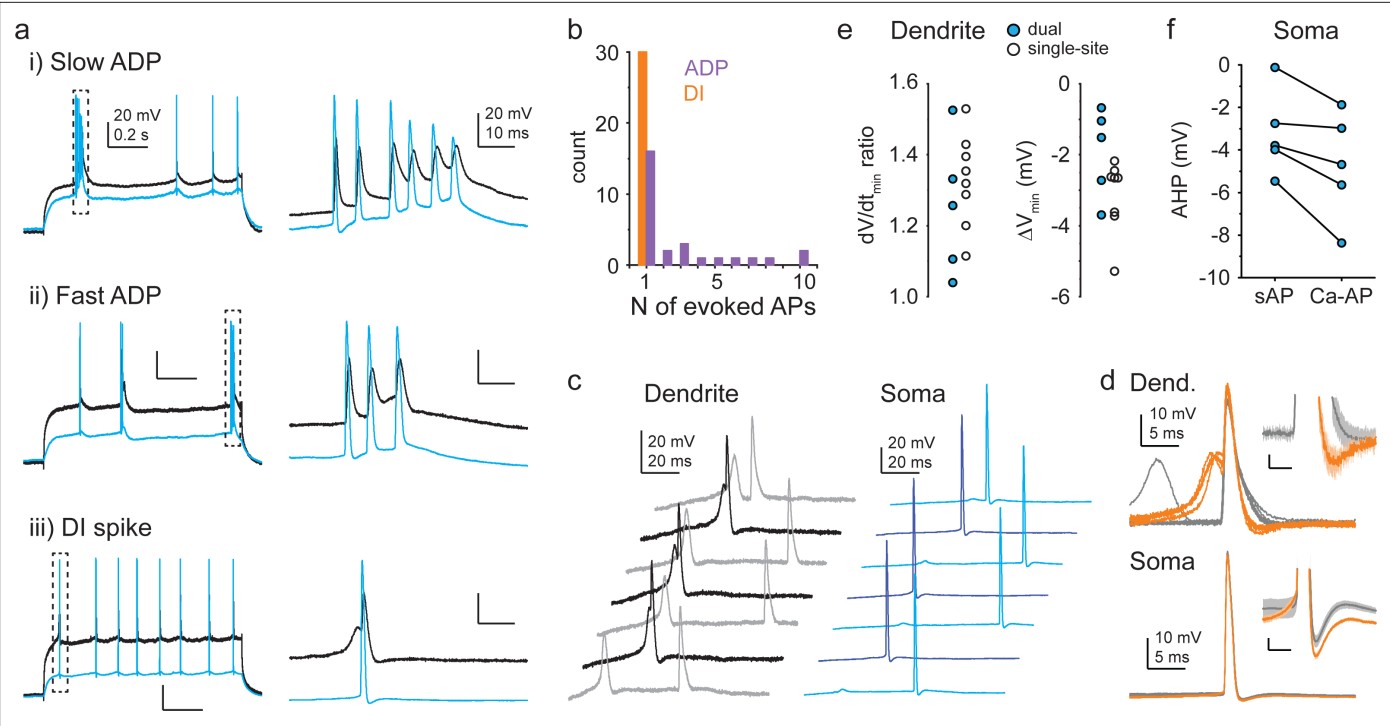

**Figure 8.** Opposing impact of different Ca²⁺ spike types on somatic output. (**a**) Example of action potential (AP) output evoked by slow afterdepolarization (ADP) (top), fast ADP (middle), and dendritically initiated (DI) spike (bottom) in dual recordings (three different cells). (**b**) Median number of APs evoked at threshold I$_{inj}$ by ADPs and DI spikes in individual cells. Note that n = 1 AP evoked by an ADP results in a burst of two APs together with the AP initiating the ADP. Only cells with at least three Ca²⁺ spike events evoking APs were included in the analysis. (**c**) Example traces from a dual recording. Darker colors indicate traces with Ca-APs. (**d**) Ca-APs (orange) and s-APs (gray) aligned to peak from the experiment shown in (**c**). Insets show afterhyperpolarizations (AHPs) enlarged; scale bars: 1 mV, 3 ms. (**e**) Ratio of repolarization rate (dV/dt$_{min}$, left) and difference in minimum V$_m$ within 12 ms after peak (right) of Ca-APs and s-APs measured in dendrites. Blue-filled circles: dual recordings; open circles: dendrite-only recordings. (**f**) Somatic AHP amplitude following Ca-APs and s-APs in dual recordings.

The online version of this article includes the following figure supplement(s) for figure 8:

**Source data 1.** Opposing impact of different Ca²⁺ spike types on somatic output.

**Figure supplement 1.** Effect of dendritically initiated (DI) spikes on action potential (AP) output in single-site dendritic recordings.

## Distinct Ca²⁺ spike types have opposing impact on the form of somatic output

How do the diverse dendritic Ca²⁺ spike types influence the somatic output of CA3PCs? The widely accepted idea is that the primary effect of Ca²⁺ spikes on output is the generation of bursts. ADP-type Ca²⁺ spikes indeed served such a role: they often triggered additional somatic spike(s) to produce short (2–3 APs) or long (>3 APs) series of APs or bursts after the first initiating AP (*Figure 8a and b*). In contrast, DI spikes never evoked more than a single somatic AP (*Figure 8a–c*). Furthermore, compared to simple bAPs (s-APs), the bAP evoked by DI spikes (Ca-APs) was followed by a faster repolarization rate (ratio of dV/dt$_{min}$ (Ca-APs/s-APs): 1.29 ± 0.04, p=0.001, Wilcoxon test compared to 1) and enhanced fast afterhyperpolarization (difference of AHP V$_{min}$ (Ca-AP – s-AP): –2.68 ± 0.34 mV, p=0.001, Wilcoxon test compared to 0) in the dendrites (n = 13, pooled dual [n = 5] and dendrite-only [n = 8] recordings, *Figure 8c–e*, *Figure 8—figure supplement 1*) as well as larger AHP at the soma (s-AP: –3.21 ± 0.89 mV, Ca-AP: –4.70 ± 1.12 mV, measured in n = 5 dual recordings, p=0.043, Wilcoxon test, *Figure 8c, d and f*). Thus, the impact of DI spikes on somatic output is the opposite of that classically proposed for dendritic Ca²⁺ spikes: they not only trigger strictly single APs, but they even suppress the probability of consecutive AP generation and thereby actively reduce burst output.

## Discussion

Using dual soma-dendritic and single-site dendritic patch-clamp recordings combined with 2P Ca²⁺ imaging, we reveal complex active properties of CA3PC apical dendrites. Besides supporting local Na⁺ and NMDAR-mediated d-spikes (*Brandalise and Gerber, 2014*; *Kim et al., 2012*; *Makara and Magee, 2013*), we report that these dendrites express distinct types of Ca²⁺ spikes that can oppositely impact the form of firing by the neuron.

One prominent component is provided by slow Ca²⁺ spikes, which are primarily responsible for generating ADP following initial AP(s) and evoke bursts of additional APs at the soma. While ADP-type Ca²⁺ spikes have been described in apical trunks of various PC types, their properties have rarely been systematically evaluated and compared. We found that the slow Ca²⁺ spikes underlying ADPs in CA3PCs are typically longer lasting (on average approximately threefold) than similar type Ca²⁺ spikes in CA1PC trunks. We also showed that slow Ca²⁺ spikes appear globally throughout the apical (or even the whole) dendritic arbor, and they are most prevalent in CA3PCs with relatively long single primary trunks. These properties altogether well align with the high burst propensity observed in deep distal CA3PCs, which show this morphological feature (*Hunt et al., 2018*; *Raus Balind et al., 2019*). At this point, it is not clear whether slow Ca²⁺ spikes are evoked *ab ovo* as a 'global' dendritic spike (*Connelly et al., 2015*) or they have a specific generation zone from where they invade the arbor. The results suggest that the long primary apical trunk acts as a designated generation site for slow Ca²⁺ spikes, but it is also possible that this dendritic trait simply correlates with other features (possibly including passive and/or active ion channel-mediated dendritic properties) that promote slow spike generation.

In addition to slow Ca²⁺ spikes, depolarization also evoked a novel fast form of DI spikes in a large fraction of CA3PC trunk dendrites. DI spikes are generated by fast Ca²⁺ spikes, which, if large enough, can recruit subsequent contribution by regenerative VGNC activation. Especially when combined into such hybrid d-spike, fast Ca²⁺ spikes can produce large enough depolarization in a window above threshold stimulation to efficiently evoke a tightly coupled AP at the soma. Curiously, these APs are immediately followed by enhanced repolarization and AHP both at dendrite and soma, cutting off depolarization and actively blocking burst firing. As a result, DI spikes produced strictly single APs, a form of input-output transformation that is the opposite of that generally proposed for Ca²⁺ spikes. This means that the regular spiking firing profile of certain CA3PCs may be not simply due to a lack of d-spikes (or other mechanisms) promoting bursts, but instead due to a specific form of local dendritic spikes that actively produces non-bursting, regular spiking firing phenotype. From another point of view, DI spikes might serve to amplify certain local apical synaptic input forms without strong peri-somatic dendritic activity (i.e., no bAPs) to promote an output, yet they do so in a way that prevents switching to burst firing, which can be preserved for signaling different input patterns, for example, associative input conjunction (*Raus Balind et al., 2019*).

Another interesting aspect of DI spikes is their semi-compartmentalization: they propagate well towards distally in the dendritic subtree belonging to the depolarized intermediary trunk, but they propagate poorly proximally and fail near the major bifurcation zone ~100 μm from the soma. Thus, dendritic families composed of higher-order trunks with connected daughter branches can correspond to independent spatial units of synaptic integration and/or plasticity (~3–5 units per cell). This represents a 'mesoscale' level of compartmentalization that is intermediate between global and local (branch specific) d-spikes. One might speculate that DI spikes could be particularly suited to promote compartmentalized induction of synaptic or intrinsic (*Losonczy et al., 2008*) plasticity because the evoked single APs, backpropagating from the soma to other parts of the dendritic arbor, would likely be inefficient to induce substantial plasticity outside the spike-generating dendritic family. Such compartmentalized plasticity could lead to biased wiring of correlated synaptic inputs targeting specific apical subtrees. It remains to be seen whether such mesoscale connectivity structure exists in CA3PCs, in addition to the fine-scale clustering of inputs previously shown (*Takahashi et al., 2012*). Notably, due to their strong attenuation, DI spikes are challenging to detect and likely remain over-looked with somatic voltage recording techniques; multisite measurements in dendrites and soma, for example, with fast voltage imaging techniques will be required to uncover this form of input-output transformation in vivo.

While we classified the dendrites based on whether they expressed DI spikes, it is likely that large transient ADPs that we also observed in some of the dendrites were mediated by a similar fast Ca²⁺ spike mechanism to that producing DI spikes. In fact, in several dendrites smaller $I_{inj}$

evoked fast ADPs, whereas at higher $I_{inj}$ DI spikes occurred. We speculate that at weaker depolarization the arrival of a bAP can provide the necessary stimulus to trigger the $Ca^{2+}$ spike that therefore appears as an ADP; the inactivation of VGNCs during the initializing bAP may prevent their further contribution to the spike. In contrast, stronger depolarization can itself initiate the fast $Ca^{2+}$ spikes, which in turn can trigger an additional $Na^+$ spike component to develop the full fast hybrid DI spike. Notably, a substantial fraction of CA3PCs coexpressed different types of $Ca^{2+}$ spikes. Altogether, the various forms of fast and slow dendritic $Ca^{2+}$ spikes in the complex branching apical tree provide particularly large room for specific input-output transformations in CA3PCs depending on the strength and actual spatiotemporal pattern of activity of the three different sources of afferent synaptic inputs.

Further studies are required to elucidate the biophysical mechanisms and implications of the diverse spike characteristics. In particular, future work is needed to resolve whether the diversity may be explained by differences in the molecular composition or functional properties of various VGCC subtypes producing kinetically different $Ca^{2+}$ currents (*Avery and Johnston, 1996*) or other, particularly $K^+$ channel types (e.g., specific voltage-dependent or $Ca^{2+}$-activated $K^+$ channels), as well as the role of passive properties. While ion channels may be regulated in an activity- or state-dependent fashion, the observed correlation of functional and morphological properties of CA3PCs suggests an at least partially rigid organization of d-spike mechanisms in different dendritic compartments (although proximal apical CA3 dendritic morphology is plastic; *Juraska et al., 1989*; *McEwen and Magarinos, 1997*). A reasonably long primary trunk appears to allow the generation of slow $Ca^{2+}$ spikes with a low threshold that can be reached both by synaptic/dendritic and somatic depolarization (*Raus Balind et al., 2019*), whereas higher-order trunks may be diverse individual $Ca^{2+}$ spike generators. It needs to be studied by future work how the summation of the various d-spikes at the soma finally determines the shape and burstiness of AP output. It will be also interesting to explore the theoretical aspects of how the complex rules driving specific output responses and plasticity mechanisms evoked by different state-dependent input combinations can contribute to the postulated pattern separation and pattern completion functions of the CA3 network during spatial navigation and memory processes.

The functional properties of DI spikes we discovered in rat CA3PCs resemble those recently described for dendritic $Ca^{2+}$ spikes (called dCaAPs) in L2/3PCs of the human neocortex derived from patients operated for epilepsy or tumor (*Gidon et al., 2020*). Except for somewhat different amplitudes, the kinetics and the inverse relationship of amplitude with depolarization of these spikes in rat CA3PCs was similar to that in hL2/3PCs. While dCaAPs of human neurons were present in TTX (similar to our results), comparison of spike properties before and after TTX application was not reported; therefore, it remains to be determined whether dCaAPs in human neurons have similar hybrid $Ca^{2+}$/ $Na^+$ spike components as DI spikes in rat CA3PCs. Altogether, our results contradict the idea that DI $Ca^{2+}$ spikes would have developed to support human specific cortical computations. Instead, it is intriguing to speculate that DI spikes may contribute to basic circuit computation motifs shared by hippocampal CA3 and superficial cortical PCs, for example, related to their intermediate position in a sequential chain of input processing (*Shepherd, 2011*). Finally, the presence of DI $Ca^{2+}$ spikes in neurons of healthy rats is a strong indication for their physiological role and argues against a possibility that the expression of these spikes in human neurons could have been related to disease or pharmacological treatment.

Altogether, our results point out that different types of PCs may utilize different structure-function models for dendritic computations (i.e., different architecture of hierarchical and parallel nodes of nonlinear synaptic integration) that allow complex forms of input-output transformations. The cell-type-specific forms of dendritic spikes raise the idea that active dendrites may support circuit-specific computations, whereas the heterogeneity within a principal cell class suggests that PC subpopulations may be dedicated to perform different information processing functions. Elucidating how these diverse models are employed in vivo under behaviorally relevant conditions of natural complex excitatory input patterns, inhibition, and neuromodulation (*Sheffield et al., 2017*; *Takahashi et al., 2020*) will be a major step towards understanding how single-cell computations can serve the network functions underlying appropriate and flexible behaviors to adapt to environmental challenges.

## Materials and methods

### Hippocampal slice preparation

Adult male Wistar rats (7–12 weeks old, RRID:RGD_737929) were used to prepare 400-μm-thick slices from the hippocampus as described (*Makara and Magee, 2013*; *Raus Balind et al., 2019*), according to methods approved by the Animal Care and Use Committee of the Institute of Experimental Medicine, and in accordance with the Institutional Ethical Codex, Hungarian Act of Animal Care and Experimentation 40/2013 (II.14), and European Union guidelines (86/609/EEC/2 and 2010/63/EU Directives). Animals were deeply anesthetized with 5% isoflurane and quickly perfused through the heart with ice-cold cutting solution containing (in mM): sucrose 220, $NaHCO_3$ 28, KCl 2.5, $NaH_2PO_4$ 1.25, $CaCl_2$ 0.5, $MgCl_2$ 7, glucose 7, Na-pyruvate 3, and ascorbic acid 1, saturated with 95% $O_2$ and 5% $CO_2$. The brain was quickly removed and slices were prepared in cutting solution using a vibratome (VT1000S, Leica, Leica Biosystems GmbH, Nussloch, Germany). Slices were incubated in a submerged holding chamber in ACSF at 35°C for 30 min and then stored in the same chamber at room temperature.

### Patch-clamp recordings

Slices were transferred to a custom-made submerged recording chamber under the microscope where experiments were performed at 32–34°C in ACSF containing (in mM) NaCl 125, KCl 3, $NaHCO_3$ 25, $NaH_2PO_4$ 1.25, $CaCl_2$ 1.3, $MgCl_2$ 1, glucose 25, Na-pyruvate 3, and ascorbic acid 1, saturated with 95% $O_2$ and 5% $CO_2$. Neuron were visualized using Zeiss Axio Examiner or Olympus BX-61 epifluorescent microscope with water immersion lens (63× or 60× during recording, 20× or 10× for overview z-stacks, Zeiss or Olympus). Higher-order apical dendritic trunks in str. radiatum were patched under oblique or Dodt contrast illumination. After establishing the dendritic whole-cell configuration, cells were loaded for >10 min to visualize the soma and dendritic tree by 2P imaging. Neurons were carefully inspected for TEs on thick parent dendrites (to verify they were CA3PCs), and that no main dendritic trunk was cut. In dual recordings, the soma was patched subsequently, guided by 2P imaging. Neurons included in this study were all recorded in different slices, typically from different animals.

Dendritic (6–10 MΩ) and somatic (4–6 MΩ) patch pipettes were filled with a solution containing (in mM) K-gluconate 134, KCl 6, HEPES 10, NaCl 4, $Mg_2ATP$ 4, $Tris_2GTP$ 0.3, phosphocreatine 14 (pH = 7.25) complemented with 50 μM Alexa Fluor 594 and 100 μM Oregon Green BAPTA-1 (OGB-1) or Oregon Green BAPTA-6F (OGB-6F) (all fluorescent dyes from Invitrogen-Molecular Probes). Electrophysiological results were similar using OGB-1 and OGB-6F, and therefore results obtained with different $Ca^{2+}$-sensitive dyes were pooled.

Current-clamp whole-cell recordings were performed using BVC-700A amplifiers (Dagan, Minneapolis, MN) in the active 'bridge' mode, filtered at 3 kHz and digitized at 50 kHz. Series resistance was typically between 15 and 30 MΩ at the soma and 25–60 MΩ at the dendrite (if possible, reduced by gently blowing into the pipette), frequently checked and compensated with bridge balance and capacitance compensation. Membrane potentials ($V_m$) are reported without correction for liquid junction potential (~10 mV). The baseline $V_m$ was kept at –68 to –72 mV with appropriate constant current injection. Dendritic $V_m$ of CA3PCs ranged between –53 and –73 mV, $R_{in}$ was 105 ± 3 MΩ. In all dual recordings, somatic $V_m$ was more negative than –60 mV (*Raus Balind et al., 2019*). A set of experiments were performed on apical trunk dendrites of CA1PCs at ~230–460 μm dendritic distance from the soma; experiments were identical to those in CA3PCs except that $V_m$ was held between –64 and –68 mV. Dendritic $V_m$ of CA1PCs ranged between –59 and –62 mV, and $R_{in}$ was 44 ± 9 MΩ.

TTX, NBQX disodium salt, D-AP5 (Tocris), and $NiCl_2$ (Sigma) were prepared in stock solution in distilled water, stored at –20°C ($NiCl_2$ at room temperature) and dissolved to final concentration into bubbled ACSF before application. To reduce extracellular $Na^+$ concentration, in some experiments NaCl was replaced with equal concentration of NMDGCl in the ACSF. Modified ACSF solutions were applied for at least 10 min before testing their effect.

### Two-photon imaging and uncaging

A dual galvanometer-based two-photon scanning system (Bruker, formerly Prairie Technologies, Middleton, WI) was used to image the patched neurons and to uncage glutamate at individual dendritic spines as previously described (*Makara and Magee, 2013*; *Raus Balind et al., 2019*). Two ultrafast pulsed laser beams (Chameleon Ultra II; Coherent, Auburn, CA) were used: one laser at 920 or 860 nm for imaging OGB dyes and Alexa Fluor 594, respectively, and the other laser tuned

to 720 nm to photolyze MNI-caged-L-glutamate (Tocris; 10 mM in ACSF) that was applied through a puffer pipette with an ~20–30 μm-diameter, downward-tilted aperture above the slice using a pneumatic ejection system (PDES-02TX, NPI, Tamm, Germany). The intensity of the laser beams was independently controlled with electro-optical modulators (model 350-80, Conoptics, Danbury, CT). Linescan $Ca^{2+}$ measurements were performed with 6–8.8 μs dwell time at ~200–300 Hz.

Glutamate uncaging was performed at a clustered set of 20 spines on the patched apical trunk using 0.5 ms uncaging duration at each spine with 0.1 ms intervals between synapses, repeated five times at 40 Hz (i.e., gamma burst stimulus) (*Raus Balind et al., 2019*). Uncaging laser power was adjusted to yield compound voltage responses near the threshold of regenerative events (bAPs or dendritic spikes), preferably so that both subthreshold and suprathreshold responses could be evoked. The average amplitude of the voltage response at the first pulse was 19.2 ± 2 mV (n = 10 dendrites).

## Data analysis

Analysis of voltage and $Ca^{2+}$ recordings was performed using custom-written macros in IgorPro (Wave-Metrics, Lake Oswego, OR). For calculating input resistance and soma-dendrite voltage transfer, we measured the steady-state voltage change to 50 pA, 300 ms hyperpolarizing step current injections from baseline $V_m$. In cells with a high rate of spontaneous excitatory postsynaptic potentials (EPSPs), the above electrophysiological properties were not determined.

$Ca^{2+}$ signals are expressed as $\Delta F/F_0 = (F(t)-F_0)/F_0$, where $F(t)$ is fluorescence at a given time point and $F_0$ is the mean fluorescence during 50 ms preceding the depolarizing $I_{inj}$. To measure $Ca^{2+}$ spike-associated $Ca^{2+}$ signal amplitude (*Figure 6*), traces were aligned to the initial rise (300–400 mV/s dV/dt value) of the $Ca^{2+}$ spike, and we calculated the difference between the maximum average of 10 (OGB-1) or 3 (OGB-6F) consecutive points following the spike and the average of the 20–50 ms period preceding the spike. Note that this measurement may slightly overestimate the amplitude. $Ca^{2+}$ traces and some of the electrophysiological traces were slightly smoothed (binomial filter, N = 1) for display purposes only.

$Ca^{2+}$ spike properties were quantified using $I_{inj}$ levels at or 100 pA above the lowest $I_{inj}$ evoking the spike, unless otherwise indicated. Usually 5–10 traces were recorded with each $I_{inj}$ level, and data were averaged. Where multiple $Ca^{2+}$ spikes occurred within a trace, we analyzed the one with the largest amplitude, which was most often the first event.

ADPs were defined as regenerative depolarization initiated by one or two preceding APs. Fast ADP amplitude was measured on events that showed additional rising phase and clear peak following the initiating AP(s). We avoided ADP amplitude measurement in cases where the peak could not be defined (such as with constant decay of voltage or due to the generation of a consecutive AP) and when the ADP-evoking AP was preceded by >2 APs in the burst. ADP peak amplitude was measured from the $V_m$ immediately preceding the first AP initiating the ADP. We note that this measurement likely overestimates the true $Ca^{2+}$ spike amplitude due to the tail of the decaying AP on which the $Ca^{2+}$ spike is riding. Slow ADPs did not have a clear peak; amplitude was measured as the maximum sustained depolarization between consecutive APs in the burst compared to the $V_m$ immediately preceding the first initiating AP. Width at half amplitude of ADPs was not determined due to the uncertain rise and amplitude.

DI spikes were defined as regenerative events that were not initiated by APs. Peak amplitude of the DI spike was determined in cases where voltage reached a clear peak (either alone or before evoking a consecutive AP). The threshold of DI spikes was measured at the inflection point where voltage began to deviate from subthreshold baseline depolarization. This was sometimes difficult to determine, and when possible (at $I_{inj}$ experiments) it was aided by extrapolating to the baseline where voltage returned to after the spike, or by fitting a double exponential to the initial subthreshold voltage response to the step current injection. Width (at half amplitude) of DI spikes was measured on smoothed traces (binomial filter, N = 10) on spikes with at least 5 mV amplitude and when no APs were evoked. The parameters of pharmacologically isolated $Ca^{2+}$ spikes (measured in TTX or NMDG) were determined the same way as those of DI spikes.

The dV/dt ratio, used to support distinguishing $Ca^{2+}$ spikes and simple APs (*Figure 1—figure supplement 1*), was defined for each dendritic regenerative event as $dV/dt_{pre}/dV/dt_{peak}$, where $dV/dt_{peak}$ is the maximum dV/dt value during the event, and $dV/dt_{pre}$ is the maximum dV/dt value in the 1.5–9 ms time window preceding $dV/dt_{peak}$.

To quantify the number of evoked APs by suprathreshold $Ca^{2+}$ spikes (*Figure 8b*), results obtained at the threshold level of $I_{inj}$ (and if not enough events, at threshold +100 pA) evoking $Ca^{2+}$ spikes were analyzed, in those experiments where at least three $Ca^{2+}$ spike events (ADP or DI spike) with evoked APs were recorded. DI spikes were included in the analysis only if they were clearly separable from the evoked AP. ADP events were analyzed only after steady-state voltage was reached by $I_{inj}$. APs following ADPs were considered to be evoked by the $Ca^{2+}$ spike if they occurred before the depolarized $V_m$ returned to 3.5 mV above the baseline but maximum within 300 ms time window after the first AP, with <100 ms preceding interspike interval.

When measuring the impact of DI spikes on the AHP (*Figure 8*), in order to ensure that temporal voltage changes did not affect our comparison, we restricted our analysis to those cells where both $Ca^{2+}$ spike-evoked APs ($AP_{Ca}$) and simple APs ($AP_s$) were evoked at the same $I_{inj}$ level, either within the same traces at the steady-state depolarization phase or on intermittent traces within the same time window.

To calculate the propensity of dendritic spikes evoked by 2PGU, traces with moderate stimulus strength were used, that is, where regenerative events (bAPs and/or d-spikes) were evoked only on any of the last three pulses of the gamma burst. The relative ADP and DI spike probability (range: 0–1) was calculated by dividing the number of traces displaying the respective d-spike type by the total number of suprathreshold traces (minimum n = 3).

Transfer of steady-state voltage signals in dual recordings was determined using hyperpolarizing current injection either to the soma or dendrite, and calculating the ratio of the resulting voltage deflection at the two locations. Attenuation of dendritic spikes was measured as the ratio of amplitudes at the dendrite and the soma. The somatic peak of fast ADPs was in some cases masked by the AHP of the AP; these cases were not included in the analysis.

We note that the number of measurements varies for different parameters due to the specific criteria applied for their analysis.

## Morphological analysis

Alexa Fluor 594 fluorescence was used for morphological analysis. Dendritic morphological and distance measurements were performed using ImageJ (NIH, Bethesda, MD) on stacked images collected at the end of the experiment. Dendritic length and distance were measured on the collapsed 2P stacks by manually drawing a segmented line along the dendrite. TEs were identified as large, lobular, complex spine-like postsynaptic structures on first- or higher-order trunks near the soma (*Chicurel and Harris, 1992*; *Raus Balind et al., 2019*). The total coverage of dendrites by TEs was estimated as the sum of the length of freehand lines drawn manually along the dendritic segments where TEs were observed.

## Statistical analysis

No statistical methods were used to predetermine sample sizes, but our samples are similar to or exceed those reported in previous publications and that are generally employed in the field. Statistical analysis was performed with the Statistica software (Statsoft, Tulsa, OK). Usually nonparametric tests (Wilcoxon test for two paired groups, Mann–Whitney test for two unpaired groups, Spearman correlation) were used, which do not make assumptions about the distribution of data. For data with mixed-factor design, two-way repeated measures ANOVA test was used with Tukey's test for post hoc comparisons; in these analyses, all data passed the Levene test. All statistical tests were two-tailed. Differences were considered significant when $p < 0.05$. In all figures, population data are represented by symbols and error bars showing mean ± SEM: *$p < 0.05$; **$p < 0.01$; ***$p < 0.001$.

## Acknowledgements

We thank Z Varga-Németh for excellent technical assistance, and BB Ujfalussy and Z Nusser for helpful discussions and comments on the manuscript. This work was supported by the European Research Council (ERC) under the European Union's Horizon 2020 research and innovation programme (CoG, grant agreement No 771849 to JKM), the International Research Scholar Program of the Howard Hughes Medical Institute (55008740 to JKM), and the NKFIH (K-124824 to JKM).

## Additional information

### Funding

| Funder | Grant reference number | Author |
|---|---|---|
| European Research Council | ERC-CoG 771849 | Judit K Makara |
| Howard Hughes Medical Institute | 55008740 | Judit K Makara |
| National Research Development and Innovation Office | K-124824 | Judit K Makara |

The funders had no role in study design, data collection and interpretation, or the decision to submit the work for publication.

### Author contributions

Ádám Magó, Formal analysis, Investigation, Methodology, Writing - original draft, Writing - review and editing; Noémi Kis, Investigation, Writing - review and editing; Balázs Lükő, Software; Judit K Makara, Conceptualization, Formal analysis, Funding acquisition, Investigation, Project administration, Resources, Supervision, Visualization, Writing - original draft, Writing - review and editing

### Author ORCIDs

Noémi Kis http://orcid.org/0000-0002-2264-3061
Judit K Makara http://orcid.org/0000-0001-8134-6334

### Ethics

This study was performed according to animal protocols approved by the Animal Care and Use Committee of the Institute of Experimental Medicine, and in accordance with the Institutional Ethical Codex, Hungarian Act of Animal Care and Experimentation 40/2013 (II.14), and European Union guidelines (86/609/EEC/2 and 2010/63/EU Directives).

### Decision letter and Author response

Decision letter https://doi.org/10.7554/eLife.74493.sa1
Author response https://doi.org/10.7554/eLife.74493.sa2

---

## Additional files

### Supplementary files

• Transparent reporting form

### Data availability

Data generated and analysed during the study are included in the manuscript and supporting files. Source data are provided for all relevant figures.

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
