## [Editor Report]

In this technically challenging study of CA3 pyramidal neurons in adult rats, involving dendritic patch-clamp electrophysiology from thin distal dendrites, the authors systematically and rigorously characterize dendritic calcium spikes and their heterogeneities in this neuronal subtype. The authors used dual somatic-dendritic recording, two-photon calcium imaging, and targeted glutamate uncaging to explore the active properties of dendrites of CA3 pyramidal neurons from rat hippocampus. They discovered two fundamentally different types of regenerative calcium events or spikes in the dendrites of these cells. One remained fairly local while the other propagated widely throughout the dendrites. The authors conclude that these different calcium spikes contribute to various single neuron computations. The authors rightly emphasize cell-type dependence of dendritic physiology across different neuronal subtypes and underscore the need to account for these differences in understanding single-neuron and circuit physiology. Overall, this interesting, important, and well-done study delineates the different dendritic calcium events in CA3 pyramidal neurons employing various kinds of stimuli.

---

## [Decision Letter]

[Editors’ note: the authors submitted for reconsideration following the decision after peer review. What follows is the decision letter after the first round of review.]

Thank you for submitting the paper "Distinct dendritic ca^2+^ spike forms produce opposing input-output transformations in rat CA3 pyramidal cells" for consideration by *eLife*. Your article has been reviewed by 2 peer reviewers, including Rishikesh Narayanan as the Reviewing Editor and Reviewer #1, and the evaluation has been overseen by a Senior Editor.

We are sorry to say that, after consultation with the reviewers, we have decided that this work, as it currently stands, will not be considered for publication by *eLife*. However, we encourage resubmission to *eLife* if the following essential experiments are performed, and the results are incorporated into a revised version of the manuscript.

As you would see from reviewers' comments, both reviewers were positive about the rigor of the study and the conceptual advances presented in the manuscript. They were appreciative of the technical challenges that the CA3 presents, and the utility of the data presented in the manuscript. However, as there are essential experiments required to support the physiological relevance of the conclusions here, the decision was to reject the manuscript and encourage resubmission if these additional essential experiments are carried out. Based on consultations, the following were deemed essential for a potential resubmission (with new data required in addressing Pt. 1 below).

1. The 1-sec step current injection is a very unphysiological stimulus. Substantiate your conclusions with more physiological forms of stimulation such as synaptic stimulation, glutamate uncaging, or dynamic clamp (that simulates the conductance changes from single or trains of synaptic input). These experiments would test whether the calcium events were dependent on the unphysiological long-current injection protocol.

2. Discuss in detail the potential biophysical mechanisms towards providing explanations for the different calcium events, especially focusing on different subtypes of calcium channels and transient potassium channels. Account for new data from Pt. 1 in refining such explanations. Additional pharmacological data on the impact of these ion channels on the different calcium events would provide crucial insights on mechanistic origins, but such data is not deemed as part of essential revisions.

3. Refine the nomenclatures associated with the two types of calcium events, specifically accounting for existing nomenclatures on dendritic spikes across different cell types.

4. Make changes to the manuscript accounting for all comments by the reviewers in a point-by-point fashion.

*Reviewer #1:*

In this technically challenging study, involving dendritic patch-clamp electrophysiology from thin distal dendrites of CA3 pyramidal neurons in adult rats, the authors systematically and rigorously characterize dendritic calcium spikes and their heterogeneities in this neuronal subtype. The authors convincingly distinguish dendritic sodium spikes from the calcium spikes that they describe in this study. They classify dendritically initiated calcium spikes into those that follow axo-somatic spikes (ADP-type) and those that are 'independently' initiated in the dendrites. They demonstrate differences in spatiotemporal spread of ADP-type vs. independent dendritic calcium spike events, with independent spikes showing fast kinetics with enhanced compartmentalization. They demonstrate that a fraction dendritically initiated independent calcium spikes result in single axo-somatic spikes, and not an axo-somatic burst that is typically associated with slow dendritic calcium events. The authors rightly emphasize cell-type dependence of dendritic physiology across different neuronal subtypes, and underscore the need to account for these differences in understanding single-neuron physiology, by employing dendritic recordings. I believe that this systematic characterization study provides novel insights into CA3 dendritic physiology, and into the comparative physiology across dendrites in different neuronal subtypes, and is complete by itself as an important stand-alone contribution to the neuroscience literature.

1. Important: I presume that the recordings reported here were performed in the presence of synaptic receptor blockers. The ACSF composition doesn't seem to report this. Please state this. This is important because if synaptic receptors weren't blocked some of the interpretations might have to be revisited given the strong recurrent circuitry and spontaneous activity in the CA3 circuit.

2. I am not entirely convinced with the nomenclature of 'independent' spikes. Across neurons, it is well-established that dendritic spikes need not always result in axo-somatic action potentials (Golding et al., Neuron, 2001; Larkum et al., Science, 2009) – irrespective of the depolarizing current (Na/Ca/NMDAR) that yields the dendritic spike. Even in vivo recordings point to a higher prevalence of dendritic spikes compared to axo-somatic spikes (e.g., Moore et al., Science, 2017). Thus, it is clear that there are dendritic spiking events that are independent of axo-somatic spiking across neuronal subtypes. Thus, this confusing new nomenclature is non-essential, and simply to the jargon. I agree that it is important to distinguish dendritically initiated events from ADP-type events that are initiated by (or require or follow) axo-somatic spikes and those that are dendritically initiated. But, traditionally the independent dendritic events have simply been referred to as dendritic spikes. The authors might want to consider this and avoid further jargon.

3. The Discussion section repeats several points. Instead the authors might want to focus on future directions, especially relating to mechanistic bases and implications associated with the findings reported here.

For instance, I believe that future follow-up studies could probe the biophysical mechanisms behind the reported physiological characteristics: (i) the dendritic electrogenesis of calcium events, and the role of the location of electrogenesis in some of the reported heterogeneities; (ii) branch-specific heterogeneities in calcium spike generation, and the role of dendritic morphology and different regenerative and restorative conductances in mediating such heterogeneities; (iii) the mechanistic basis of the spatio-temporal distinctions observed between ADP-type and independent calcium spike events, especially focusing on the contributions of cable-theoretic vs. active components in the spatiotemporal spreads of fast vs. slow events; (iv) the mechanistic basis for the larger AHP associated with independent dendritic calcium spike events and why they result in single spikes while ADP-like events result in burst, including the role of calcium-activated potassium channels that express in abundance in these cell types; (v) the mechanistic basis for the reduction in the amplitude of dendritic calcium events with depolarization, including inactivation of voltage-gated calcium and sodium channel triggered by what is traditionally referred to as depolarization-induced block of spiking (in this case dendritic spiking); and (vi) the implications of their observations to CA3 neuronal and network physiology, especially given the recurrent circuitry impinging on the SR, the detonator synapses impinging on the SL and the distal entorhinal inputs – in the context of specific postulated functions of the CA3 network in spatial navigation, pattern completion, etc.

4. For future studies, the authors might want to consider using Methyl Sulphate or Methanesulphonate as the charge-carrying anion. Gluconate acts as a weak calcium buffer, and thereby affecting normal function of calcium-activated potassium channels. Given the important role of this channel subtype in CA3 pyramidal neuron physiology, the authors might want to avoid Gluconate for future studies (at least in a subset of recordings, given instabilities associated with the charge-carrying anions other than Gluconate).

*Reviewer #2:*

This is a rigorous and comprehensive analysis of active dendrites in a poorly studied neuron, CA3 pyramidal neurons from slices of rat hippocampus. The authors used exceedingly difficult methods for their study, and the results and conclusions follow closely from their experiments. They conclude that there are two types of regenerative Ca events or spikes in the dendrites of these cells, which they call, fast "ADP" spikes and slower "independent" spikes. As expected, the slower events spread widely in dendrites while the fast events remain more local. Also, the ability to generate the slow events roughly correlates with cell morphology and the density of mossy fiber thorny excrescences. Finally, they find that the two types of events trigger somatic firing differently.

The authors used 1-sec long depolarizing current injections from either the soma or dendrites to trigger these events. The presence of dendritic Ca spikes is completely dependent on the various types of voltage-gated ion channels expressed in dendrites. These channels include the multiple types of Ca channels, fast and persistent Na channels, numerous types of K channels, and HCN channels. The authors used TTX to separate the role of Na channels in these events and a non-selective Ca channel blocker to show that the events were indeed Ca dependent, but no other pharmacology was used to explore the role of some of the other dendritic channels. For example, separating inactivating T-type and non-inactivating L-type channels in the generation of these events would have been interesting. Also, inactivating K channels (Kv4 or A-type) have been shown to play a major role in regulating regenerative events in dendrites of many cell types but the authors did not explore or mention these channels. The authors thus provide no mechanism or explanation for the different Ca events.

The 1-sec step current injection is an unphysiological stimulus. Subtle differences in Ca spikes could very well be dependent on the stimulus given and whether or not inactivation of Ca and K channels played a role in their generation. The results would have been more convincing with more physiological forms of stimulation such as synaptic stimulation, glutamate uncaging, or α-function current injection to test whether the Ca events were dependent on the long current injection.

In summary, this is an interesting, important and well-done study that is limited in its conclusions by the unphysiological stimulus used and the lack of a mechanism for the two types of Ca events.

Some of my criticisms of this paper are described above. I could not recommend publication without the authors addressing some of them. In particular, the authors must demonstrate with additional experiments that the separation of the Ca events into two modes is not dependent on the 1-sec current injection. They should use one of the methods mentioned above. Current injection waveforms that mimic synaptic conductance changes would probably be the easiest, or, even better, using a dynamic clamp that simulates the conductance changes from single or trains of synaptic input.

---

## [Author Response]

[Editors’ note: the authors resubmitted a revised version of the paper for consideration. What follows is the authors’ response to the first round of review.]

1. The 1-sec step current injection is a very unphysiological stimulus. Substantiate your conclusions with more physiological forms of stimulation such as synaptic stimulation, glutamate uncaging, or dynamic clamp (that simulates the conductance changes from single or trains of synaptic input). These experiments would test whether the calcium events were dependent on the unphysiological long-current injection protocol.

We have performed new experiments using multisite two-photon glutamate uncaging at trunk synapses near the dendritic recording pipette. We stimulated 20 spatially clustered spines quasisynchronously 5 times at 40 Hz, with laser power producing moderately suprathreshold activity (evoking bAP or d-spike on the last 3 pulses). To avoid activation of confounding slow NMDA spikes, the experiments were performed in the presence of an NMDAR blocker in the bath (D-AP5, 50 µM). The above synaptic stimulation was able to elicit characteristic and well distinguishable ADPs and independent (now termed ‘dendritically initiated’ or DI) spikes either separately (ADP only: 3/10 dendrites, DI spike only: 4/10 dendrites) or in combination (1/10 dendrites). We also found that the Ca^2+^ spike type evoked by uncaging corresponded well to that evoked by I_inj_ via the pipette. These new data are included in the revised manuscript as Figure 3k-o and described in the Results (ln. 253-269).

The results thus confirm that the Ca^2+^ spike form is an inherent property of the dendrite, and that separate Ca^2+^ spike modes can be elicited by wide range of stimuli that reach sufficiently strong local depolarization.

2. Discuss in detail the potential biophysical mechanisms towards providing explanations for the different calcium events, especially focusing on different subtypes of calcium channels and transient potassium channels. Account for new data from Pt. 1 in refining such explanations. Additional pharmacological data on the impact of these ion channels on the different calcium events would provide crucial insights on mechanistic origins, but such data is not deemed as part of essential revisions.

We now provide a more detailed discussion of these possibilities in the revised Discussion.

We fully agree that elucidating the biophysical basis of the distinct Ca^2+^ spike types will be of fundamental importance for future work. We are currently directing our efforts towards testing the passive (morphological) and active (channel mediated) factors that could contribute to generating the different forms of ca^2+^ spikes, using pharmacological and modeling tools. Since this is a follow-up work exploring various mechanisms in a wider scope, we plan to dedicate a separate publication to this study.

3. Refine the nomenclatures associated with the two types of calcium events, specifically accounting for existing nomenclatures on dendritic spikes across different cell types.

We ourselves had difficulties and debates about establishing a nomenclature. In our view, the general term ‘dendritic spike’ refers to the site where the regenerative event is generated, not the mechanism that initiates it. The aim of our nomenclature was to point out the differences between spikes that are initiated independently from the soma and those that are initiated by somatic APs. Since the Ca^2+^ spikes themselves are generated in the dendrite in both cases, in our opinion they should all be called dendritic spikes, and an additional identifier is needed for the initiation mechanism. (Further adds to the nomenclature problem if we want to include also information about whether the dendritic spike evokes somatic AP as a result – based on this aspect, dendritic spikes have been recently called ‘coupled’ and ‘isolated’ events (Gidon et al., 2020). We decided not to include this aspect in our nomenclature, since both ADPs and dendritically initiated spikes can evoke APs or remain subthreshold at the soma.)

Accepting the point that the term ‘independent spike’ was not well defined and descriptive, we instead now use the term ‘dendritically initiated (DI)’ spike for bAP-independent Ca^2+^ spikes. We hope that the Editor and Reviewers can share our point that, since the mechanism and impact of the two types of events are different, our nomenclature represents useful information rather than l’art pour l’art jargon.

4. Make changes to the manuscript accounting for all comments by the reviewers in a point-by-point fashion.

Please see below our point-by-point response detailing the changes in the manuscript reflecting to the reviewers’ comments.

Beyond the requested revisions, we have also performed additional dual experiments in order to increase the sample size in Figure 8f. With n=5 the result is now statistically significant (p<0.05). We included the additional dual experiments to the relevant datasets and updated the corresponding figures.

Reviewer #1:In this technically challenging study, involving dendritic patch-clamp electrophysiology from thin distal dendrites of CA3 pyramidal neurons in adult rats, the authors systematically and rigorously characterize dendritic calcium spikes and their heterogeneities in this neuronal subtype. The authors convincingly distinguish dendritic sodium spikes from the calcium spikes that they describe in this study. They classify dendritically initiated calcium spikes into those that follow axo-somatic spikes (ADP-type) and those that are 'independently' initiated in the dendrites. They demonstrate differences in spatiotemporal spread of ADP-type vs. independent dendritic calcium spike events, with independent spikes showing fast kinetics with enhanced compartmentalization. They demonstrate that a fraction dendritically initiated independent calcium spikes result in single axo-somatic spikes, and not an axo-somatic burst that is typically associated with slow dendritic calcium events. The authors rightly emphasize cell-type dependence of dendritic physiology across different neuronal subtypes, and underscore the need to account for these differences in understanding single-neuron physiology, by employing dendritic recordings. I believe that this systematic characterization study provides novel insights into CA3 dendritic physiology, and into the comparative physiology across dendrites in different neuronal subtypes, and is complete by itself as an important stand-alone contribution to the neuroscience literature.1. Important: I presume that the recordings reported here were performed in the presence of synaptic receptor blockers. The ACSF composition doesn't seem to report this. Please state this. This is important because if synaptic receptors weren't blocked some of the interpretations might have to be revisited given the strong recurrent circuitry and spontaneous activity in the CA3 circuit.

We performed the recordings without synaptic receptor blockers, as described in the Methods.

While we agree with the Reviewer that recurrent synaptic activity in CA3 is a potential concern (in particular in the case of ADPs that are preceded by somatic firing), several lines of evidence collectively indicate that synaptic and circuit mechanisms do not impact our results and conclusions.

Most importantly, to directly address this concern, we have performed additional dendritic recordings to test the effect of excitatory synaptic activity on the dendritic spikes evoked by our protocol. After assessing the dendritic spike profile by 1-sec-long I_inj_ under control conditions, we repeated the stimulation after bath application of a cocktail of AMPA and NMDA receptor blockers (10 µM NBQX and 50 µM D-AP5) for 10-15 minutes. We found that blockade of excitatory synaptic activity had no impact on the regenerative events (n=4 experiments), including ADP-type (n=3) and dendritically initiated (n=1) spikes. The efficacy of the blockers was confirmed by separate control experiments using glutamate uncaging. We have included these data into the revised manuscript as a new supplementary figure (Figure 3 —figure supplement 2) described in the Results (ln. 185-186).

Additional indirect arguments against synaptic activity influencing our results:

– In our acute slices, under control conditions we typically do not see signs of significant recurrent synaptic activity evoked after firing a single patched neuron. Recurrent activity manifests rather when using bulk extracellular electrical stimulation or when applying GABA receptor blockers.

– In previous work where we studied complex spike bursts mediated by dendritic Ca^2+^ spikes, we have tested the impact of the recurrent circuitry and found that the blockade of neither AMPA/NMDA receptors, nor GABAA/GABAB receptors affected CSB rate (Raus Balind et al., 2019 Nat. Comm.).

– The kinetics of Ca^2+^ spikes in TTX mirrors that observed under control conditions, also suggesting that they are intrinsically generated by the dendrite and not produced by recurrent activity.

2. I am not entirely convinced with the nomenclature of 'independent' spikes. Across neurons, it is well-established that dendritic spikes need not always result in axo-somatic action potentials (Golding et al., Neuron, 2001; Larkum et al., Science, 2009) – irrespective of the depolarizing current (Na/Ca/NMDAR) that yields the dendritic spike. Even in vivo recordings point to a higher prevalence of dendritic spikes compared to axo-somatic spikes (e.g., Moore et al., Science, 2017). Thus, it is clear that there are dendritic spiking events that are independent of axo-somatic spiking across neuronal subtypes. Thus, this confusing new nomenclature is non-essential, and simply to the jargon. I agree that it is important to distinguish dendritically initiated events from ADP-type events that are initiated by (or require or follow) axo-somatic spikes and those that are dendritically initiated. But, traditionally the independent dendritic events have simply been referred to as dendritic spikes. The authors might want to consider this and avoid further jargon.

We ourselves had difficulties and debates about establishing a nomenclature. In our view, the general term ‘dendritic spike’ refers to the site where the regenerative event is generated, not the mechanism that initiates it. Our aim with the nomenclature was to point out the differences between dendritic spikes that are initiated completely independently from the soma and those that require/follow somatic APs. Since the Ca^2+^ spikes themselves are generated in the dendrite in both cases, in our opinion they should all be called dendritic spikes, and an additional identifier is needed for the initiation mechanism. Further adds to the nomenclature problem if we want to include also information about whether the dendritic spike evokes somatic AP as a result – based on this aspect, dendritic spikes have been recently called ‘coupled’ and ‘isolated’ events (Gidon et al., 2020). We decided not to include this aspect in our nomenclature, since both ADPs and dendritically initiated spikes can evoke APs or remain subthreshold at the soma.

We hope that the Editor and Reviewers can share our point that, since the mechanism and impact of the two types of events are different, our nomenclature represents useful information rather than l’art pour l’art jargon. Yet, accepting the point that the term ‘independent spike’ was not well defined, we instead now use the term ‘dendritically initiated (DI)’ spike for these events.

3. The Discussion section repeats several points. Instead the authors might want to focus on future directions, especially relating to mechanistic bases and implications associated with the findings reported here.For instance, I believe that future follow-up studies could probe the biophysical mechanisms behind the reported physiological characteristics: (i) the dendritic electrogenesis of calcium events, and the role of the location of electrogenesis in some of the reported heterogeneities; (ii) branch-specific heterogeneities in calcium spike generation, and the role of dendritic morphology and different regenerative and restorative conductances in mediating such heterogeneities; (iii) the mechanistic basis of the spatio-temporal distinctions observed between ADP-type and independent calcium spike events, especially focusing on the contributions of cable-theoretic vs. active components in the spatiotemporal spreads of fast vs. slow events; (iv) the mechanistic basis for the larger AHP associated with independent dendritic calcium spike events and why they result in single spikes while ADP-like events result in burst, including the role of calcium-activated potassium channels that express in abundance in these cell types; (v) the mechanistic basis for the reduction in the amplitude of dendritic calcium events with depolarization, including inactivation of voltage-gated calcium and sodium channel triggered by what is traditionally referred to as depolarization-induced block of spiking (in this case dendritic spiking); and (vi) the implications of their observations to CA3 neuronal and network physiology, especially given the recurrent circuitry impinging on the SR, the detonator synapses impinging on the SL and the distal entorhinal inputs – in the context of specific postulated functions of the CA3 network in spatial navigation, pattern completion, etc.

We completely agree with the Reviewer that the above points are pressing questions to answer. While we have discussed our main questions for future directions to some extent in the original Discussion, we have now elaborated on the text to highlight them more clearly.

4. For future studies, the authors might want to consider using Methyl Sulphate or Methanesulphonate as the charge-carrying anion. Gluconate acts as a weak calcium buffer, and thereby affecting normal function of calcium-activated potassium channels. Given the important role of this channel subtype in CA3 pyramidal neuron physiology, the authors might want to avoid Gluconate for future studies (at least in a subset of recordings, given instabilities associated with the charge-carrying anions other than Gluconate).

Thank you for the suggestion; we will test methylsulphate in future dendritic experiments. We would like to note that in our previous study (Raus Balind et al., 2019 Nat Comm) we did not find differences in Ca^2+^ spike-driven complex spike burst propensity of CA3PCs using methysulphate vs gluconate in the pipette solution.

Reviewer #2:[…] Some of my criticisms of this paper are described above. I could not recommend publication without the authors addressing some of them. In particular, the authors must demonstrate with additional experiments that the separation of the Ca events into two modes is not dependent on the 1-sec current injection. They should use one of the methods mentioned above. Current injection waveforms that mimic synaptic conductance changes would probably be the easiest, or, even better, using a dynamic clamp that simulates the conductance changes from single or trains of synaptic input.

We have performed new experiments using multisite two-photon glutamate uncaging at trunk synapses near the dendritic recording pipette. We stimulated 20 spatially clustered spines quasisynchronously 5 times at 40 Hz, with laser power producing moderately suprathreshold activity (evoking bAP or d-spike on the last 3 pulses). To avoid activation of confounding slow NMDA spikes, the experiments were performed in the presence of an NMDAR blocker in the bath (D-AP5, 50 µM). The above synaptic stimulation was able to elicit characteristic and well distinguishable ADPs and independent (now termed ‘dendritically initiated’ or DI) spikes either separately (ADP only: 3/10 dendrites, DI spike only: 4/10 dendrites) or in combination (1/10 dendrites). We also found that the Ca^2+^ spike type evoked by uncaging corresponded well to that evoked by I_inj_ via the pipette. These new data are included in the revised manuscript as Figure 3k-o and described in the Results (ln. 253-269).

The results thus confirm that the Ca^2+^ spike form is an inherent property of the dendrite, and that separate Ca^2+^ spike modes can be elicited by wide range of stimuli that reach sufficiently strong local depolarization.